# Large Language Models Synergize with Automated Machine Learning

**Jinglue Xu**                                                    *jingluexu@gmail.com*
*University of Tokyo*

**Jialong Li**                                                    *lijialong@fuji.waseda.jp*
*Tokyo Institute of Technology*

**Zhen Liu**                                                    *liu-zhen@g.ecc.u-tokyo.ac.jp*
*University of Tokyo*

**Nagar Anthel Venkatesh Suryanarayanan**                      *nav-surya@g.ecc.u-tokyo.ac.jp*
*University of Tokyo*

**Guoyuan Zhou**                                                    *zhouguoyuan@webmail.hzau.edu.cn*
*Hosei University Institute of Integrated Science and Technology*

**Jia Guo**                                                    *guojia314@gmail.com*
*Hosei University Institute of Integrated Science and Technology*

**Hitoshi Iba**                                                    *iba@iba.t.u-tokyo.ac.jp*
*University of Tokyo*

**Kenji Tei**                                                    *tei@c.titech.ac.jp*
*Tokyo Institute of Technology*

**Reviewed on OpenReview:** *https://openreview.net/forum?id=RDEaIfOiJM*

## Abstract

Recently, program synthesis driven by large language models (LLMs) has become increasingly popular. However, *program synthesis for machine learning (ML) tasks* still poses significant challenges. This paper explores a novel form of program synthesis, targeting ML programs, by combining LLMs and automated machine learning (autoML). Specifically, our goal is to fully automate the generation and optimization of the code of the entire ML workflow, from data preparation to modeling and post-processing, utilizing only textual descriptions of the ML tasks. To manage the length and diversity of ML programs, we propose to break each ML program into smaller, manageable parts. Each part is generated separately by the LLM, with careful consideration of their compatibilities. To ensure compatibilities, we design a testing technique for ML programs. Unlike traditional program synthesis, which typically relies on binary evaluations (i.e., correct or incorrect), evaluating ML programs necessitates more than just binary judgments. Our approach automates the numerical evaluation and optimization of these programs, selecting the best candidates through autoML techniques. In experiments across various ML tasks, our method outperforms existing methods in 10 out of 12 tasks for generating ML programs. In addition, autoML significantly improves the performance of the generated ML programs. In experiments, given the textual task description, our method, *Text-to-ML*, generates the complete and optimized ML program in a fully autonomous process. The implementation of our method is available at `https://github.com/JLX0/llm-automl`.

# 1 Introduction

## 1.1 Background, challenges, and motivation

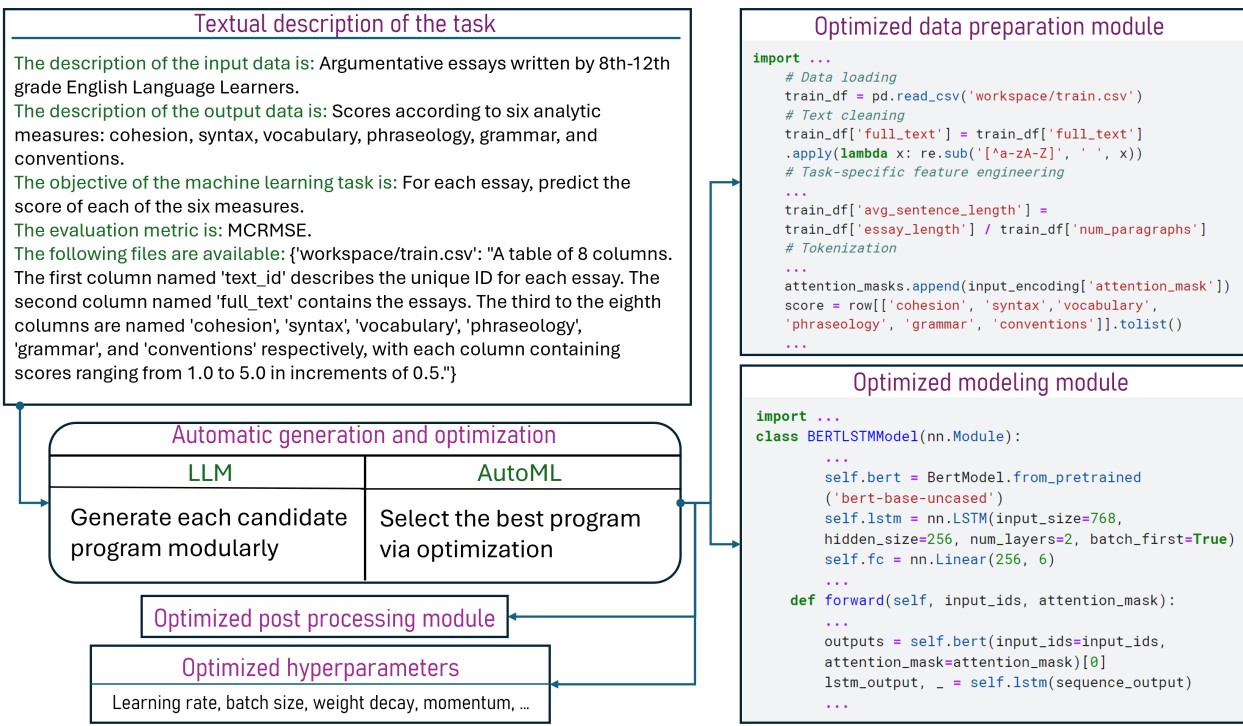

Figure 1: One-click program synthesis for machine learning. This figure presents an example of our qualitative results based on the dataset *Learning*. Our method, Text-to-ML, transforms a textual task description into an optimized program, encompassing data preparation, modeling, post processing, and hyperparameters. All modules are seamlessly integrated into a single executable program. The entire process, from the user's input of a textual description to the creation of an end-to-end, optimized, and executable program, is fully automated. The code examples in the figure are folded and abbreviated for readability.

Program synthesis is the automatic creation of computer programs that satisfy high-level specifications (Manna & Waldinger, 1971). In recent years, autoregressive large language models (LLMs) (Achiam et al., 2023; Touvron et al., 2023; Anil et al., 2023) have demonstrated impressive performance in program synthesis tasks (Austin et al., 2021; Li et al., 2022; Shinn et al., 2023). However, existing LLM-based program synthesis methods primarily focus on traditional coding problems, such as interview questions for software engineering positions (Austin et al., 2021; Chen et al., 2023b; Madaan et al., 2023), while the synthesis of machine learning (ML) programs remains largely unexplored (Figure 2). The challenges associated with program synthesis for ML stem from their length, diversity, and complexity in the testing phase:

- **Length:** The first step in our study is to generate an entire ML program, which consists of three task-specific components: (1) data preparation, such as data loading, cleaning, encoding, and feature engineering; (2) modeling, which implements an ML model, such as BERT (Devlin et al., 2018); and (3) post-processing, such as performance evaluation and decoding the results (Figure 1). These components collectively demand a lengthy output from LLMs. For instance, the validly generated code for the ML task depicted in Figure 1 consists of an average of $114 \pm 32$ lines, in stark contrast to the average $6.3 \pm 5.0$ lines of the reference solutions in the HumanEval benchmark, which addresses traditional coding problems (Chen et al., 2021). For LLMs, the output length strongly correlates with the difficulty of the generation process (Anil et al., 2022; Newman et al., 2020).

- **Diversity:** To achieve the automatic generation of the program for the entire ML program, many task-specific components of the ML program should be generated during the process, rather than relying on an inventory of pre-written modules (Surís et al., 2023; Gupta & Kembhavi, 2023; Lu et al., 2023). Depending on pre-written modules for a wide range of tasks is often impractical (Hollmann et al., 2023), especially given the vast combinations of requirements and characteristics across different ML tasks, such as the structure of dataset files, dataset features, and desired output format.

- **Complexity in testing phase:** Evaluation of programs for traditional coding problems often relies on manually written white-box tests (Chen et al., 2023b; Li et al., 2022), such as unit tests (Myers et al., 2004). Unit tests verify the correctness of individual code components, such as functions or classes, by comparing their actual behavior with the expected behavior based on specific inputs and outputs. For ML programs, unit tests are insufficient and costly to create (Tian et al., 2018; Wang et al., 2021b). From the perspective of both software testing and ML, syntactically valid ML programs should be further examined based on numerical performance metrics, such as Top-1 accuracy or mean absolute error, and a consistent evaluation procedure for semantic accuracy (Kim et al., 2019; Braiek & Khomh, 2020; Saha et al., 2022; Cambronero & Rinard, 2019).

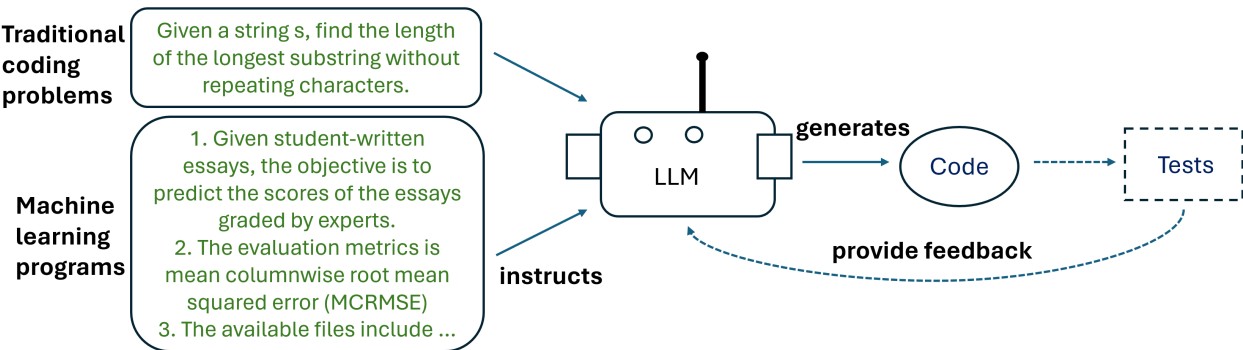

Figure 2: Program synthesis driven by autoregressive LLMs. The desired functions are provided as instructions to the LLM, which generates the corresponding code. An optional testing phase can be included, where the code is tested for errors, and the test results provide feedback for the LLM to debug and refine the code.

Notably, the complexity in testing phase entails a novel form of program synthesis, setting it apart from traditional program synthesis. In traditional contexts, program synthesis often concludes with code generation and perhaps testing through binary evaluation procedures, such as unit tests. However, ML programs necessitate numerical evaluations, extending the synthesis process beyond just code generation (Kim et al., 2019; Braiek & Khomh, 2020; Saha et al., 2022; Cambronero & Rinard, 2019).

How to ensure that each generated program is compatible with the code of a consistent and fair training, testing, and evaluation procedure? More importantly, how to effectively utilize numerical feedback involving performance scores? A numerical optimization algorithm efficiently addresses the challenge. Consequently, *for ML programs, program synthesis often necessitates optimization.* Real et al. (2020); Saha et al. (2022); Cambronero & Rinard (2019) also explore this form of program synthesis (see Section 2.3). Similar to these studies, we also utilize automated machine learning (autoML) as the numerical optimization step. However, our approach is distinct in that *the generated programs are inherently compatible with optimization algorithms, requiring no further modification.*

ML heavily impacts modern society, with applications spanning healthcare, finance, manufacturing, and beyond. Implementing ML typically demands substantial expertise and effort, which limits ML's accessibility for many potential users. Many studies work on automating ML utilizing LLMs or autoML, however, *the full automation of ML tasks at the code level remains an unresolved challenge* (see Sections 2.2 and 2.3 and Appendix A.2). In this paper, we explore the limits of program synthesis for ML by combining LLMs

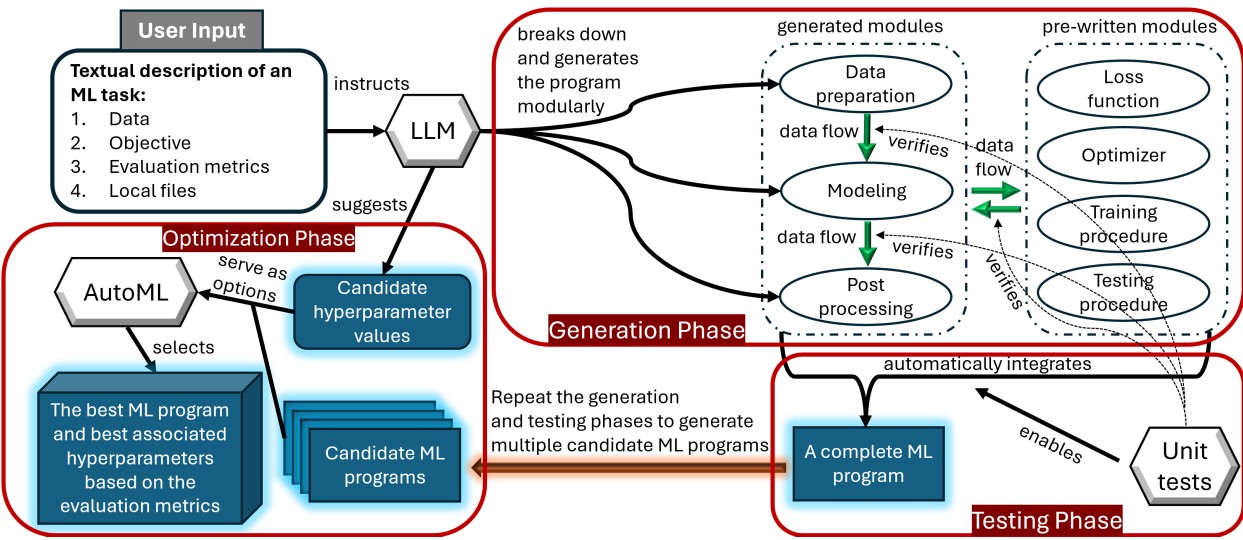

Figure 3: Text-to-ML: a three-phase process of generation, testing, and optimization. Initially, with the task description, the LLM generates the modules. The modules are verified by unit tests to ensure inter-compatibility. With the compatibility, they are automatically integrated. Then, iterate to produce multiple candidate programs. Finally, autoML selects the best program and associated hyperparameters.

and autoML. During this exploration, we aim to automate the generation and optimization of the entire ML program, reducing the human workload to mere textual descriptions of the tasks. Our study focuses on synthesizing the programs involving supervised ML tasks, encompassing traditional algorithms such as random forests and SVM, as well as deep learning domains including computer vision (CV) and natural language processing (NLP).

## 1.2 Automatic code generation

Recently, many studies have leveraged autoregressive LLMs for sequence generation tasks. For example, Chain-of-Thought (Wei et al., 2022) enhances reasoning performance by incorporating input-output examples within the input sequence, while Reflexion (Shinn et al., 2023) utilizes a reinforcement learning strategy to boost generation performance by reflecting on feedback. Despite these advancements, these frameworks still struggle with generating ML programs (see Table 1). To manage the length and diversity in ML programs, we introduce a novel sequence generation framework for autoregressive LLMs, *Contextual Modular Generation* (Algorithm 1). As a form of compositional reasoning (Andreas et al., 2016), this framework breaks down the programs into multiple smaller modules, with each module being generated in a separate process and representing a distinct code component of the overall program (Figures 1 and 3).

Unlike traditional compositional approaches (Andreas et al., 2016; Surís et al., 2023; Gupta & Kembhavi, 2023; Lu et al., 2023), Contextual Modular Generation is distinctive for two features: (1) instead of hinging on a fixed inventory of pre-written modules (e.g., selecting and reusing code from a repository of pre-written codes for available feature engineering methods), the framework dynamically generates the essential modules tailored to each specific task. In our study, the generated modules include data preparation, modeling, and post processing. (2) the generated modules are readily compatible without requiring additional sequence generation steps from LLMs for combination. This feature contributes to scalability by ensuring linear growth in generation complexity over program length (Theorem 3.8).

### 1.3 Automatic unit testing

With each module being generated in a separate process, Contextual Modular Generation requires that the generated modules be readily compatible with each other to reach the linear complexity in the generation process (Theorem 3.8). To ensure this compatibility, we devise a novel testing technique for ML programs. This technique verifies compatibility by examining that the modules exhibit consistency in data structures (such as tensor shapes) and contents (such as data features in each tensor dimension). The verification process utilizes automatically generated unit tests and synthetic data constructed from LLMs (see Section 4).

This technique not only ensures compatibility among newly generated modules but also between these modules and pre-written, less-variant modules. Specifically, the technique maintains uniformity in certain modules that should be identical across different program instances, such as the loss function, optimizer, training and testing strategies, evaluation procedure of prediction accuracy, and optimization strategy (Figure 3).

Contextual Modular Generation with the testing technique effectively generates syntactically valid ML programs (see Table 1). However, our findings align with studies such as Gu et al. (2022) and Kim et al. (2019), which suggest that unit tests are inadequate for ML applications (see Table 3). Most notably, *the semantic accuracy (i.e., the prediction performance) is often far from optimal*, which underlines the limited practical utility of ML programs generated solely by LLMs and unit tests. To address this issue, we implement autoML in pre-written modules to further evaluate and optimize the programs, capitalizing on the compatibility of generated modules with pre-written modules.

### 1.4 Automatic code optimization

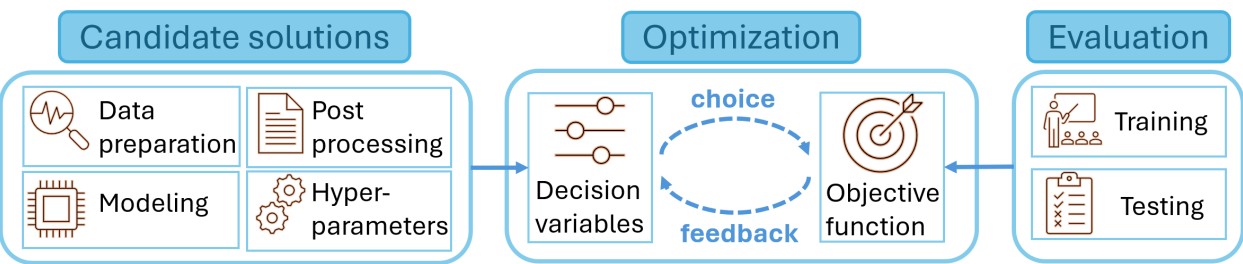

Figure 4: The autoML step in our method. Each solution is a program that implements a candidate combination of data preparation, modeling, and post-processing algorithms and associated hyperparameters. First, the optimization algorithm chooses a solution. This chosen solution is then trained and evaluated, which provides feedback to improve selections in subsequent iterations.

The last step of our method is autoML (Figure 3). AutoML is the process of automatically selecting algorithms and hyperparameters for ML tasks (Thornton et al., 2013). For a task, an autoML process begins by designating a solution and then proceeds to data processing and model training based on the solution. Subsequently, the solution is evaluated, providing feedback for future iterations to designate more promising solutions (Figure 4). Typically, autoML utilizes optimization algorithms, such as evolutionary computation or reinforcement learning, to designate the solutions. In our study, the decision variables in the optimization (i.e., the elements of the solution) include choices and hyperparameters regarding data preparation methods, ML models, and post-processing methods implemented by the ML program. The objective function (i.e., the goal of the solution) is the prediction performance of the ML program, measured by the numerical metrics.

Due to the aforementioned complexity involved in testing ML programs, existing works in program synthesis for ML often employ autoML methods (see Section 2.3). The autoML step in our method adapts the strategies in Öztürk et al. (2022) by incorporating zero-cost (ZC) proxies (Mellor et al., 2021) (see Appendix C.5). These proxies predict the performance of the trained neural networks using at most a single forward/backward propagation pass, which significantly reduces the cost from evaluating (training and testing) the models.

### 1.5 Summary

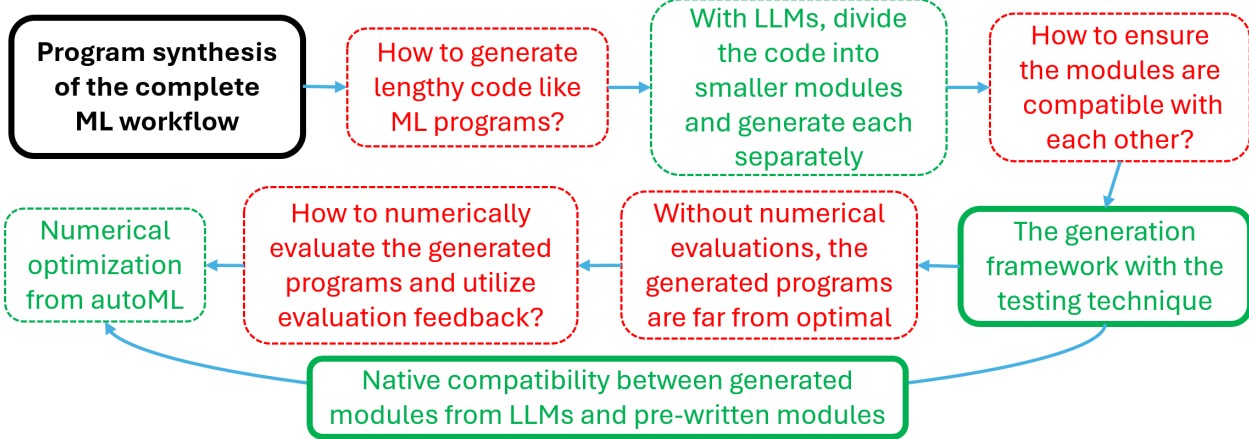

Figure 5: Program synthesis for ML. (1) black rectangle: the goal of this study. (2) red rectangles: challenges. (3) green rectangles: methods or features of methods. (4) solid rectangles: novel goals, methods, or features introduced in our study. (5) dashed rectangles: existing challenges or methods.

Figure 5 illustrates the relation among the goals, challenges, and methods in our study. Our contributions are summarized as follows:

- We explore the limits of program synthesis for ML. Qualitatively, for 12 supervised ML tasks over traditional data, CV, and NLP, *Text-to-ML* generates the code of the end-to-end, optimized program in a fully automatic process, given textual descriptions of the tasks (Figure 1 and Section 6.2). In the exploration, we offer a new perspective for combining LLMs and autoML. We identify a synergy in this combination that enhances program synthesis for ML (see Section 7.1).

- We propose a sequence generation framework *Contextual Modular Generation* for autoregressive LLMs, with a theoretical analysis of scalability over sequence length (Theorem 3.5 and Theorem 3.8). To ensure compatibility among the modules, as demanded by the framework, we devise a novel testing technique based on *progenitor testing protocols* (see Section 4) and LLMs. Our approach generates codes that are natively compatible with optimization algorithms, removing the need for extra adaptation and facilitating the immediate deployment of code optimization.

- For 10 out of 12 ML tasks across modalities, Contextual Modular Generation outperforms existing methods (Section 6.2). Furthermore, autoML substantially improves the performance of the generated programs (Section 6.4). In addition, we find that ZC proxies speed up the search with both pretrained CV and NLP models (Section 6.4).

## 2 Related Works

### 2.1 Large language models

Let the input and output sequences be $X = (x_1, x_2, \ldots, x_M)$ and $Y = (y_1, y_2, \ldots, y_N)$, where each $x_m$ or $y_n$ represents a token in the respective sequence. Let the current state of the LLM (e.g., choice of LLM, parameters in the LLM, and choice of sampling method) be $\lambda$. The conditional probability distribution of generating $Y$ given $X$ is: $p(Y|X) = \prod_{n=1}^{N} p_\lambda(y_n|y_{<n}, X)$. Throughout the paper, we denote $(a_1, a_2, \ldots, a_{N-1})$ as $a_{<N}$, where each $a_n$ is a token (and $a_{<N}$ is a sequence) or each $a_n$ is a sequence (and $a_{<N}$ is a concatenated sequence).

**Compositional reasoning** decomposes a reasoning process into multiple intermediate steps (Andreas et al., 2016). Compared to traditional compositional reasoning (Andreas et al., 2016; Surís et al., 2023; Gupta & Kembhavi, 2023; Lu et al., 2023), our method generates modules during the reasoning process. In this aspect, a similar method is Le et al. (2023). Both methods decompose the generation of the output sequence into multiple generations of smaller modules: $Y^1, Y^2, \ldots, Y^J$, where each $Y^j = (y_1^j, y_2^j, \ldots, y_k^j)$. For each $Y^j$, let the corresponding input sequence be $X^j$, then $p(Y^j|X^j) = \prod_{n=1}^N p_\lambda(y^j|y_{<n}^j, X^j)$.

A major challenge in generating the modules is whether $Y^1, Y^2, \ldots, Y^J$ form a valid solution. Le et al. (2023) addresses the issue by incorporating the modules as the input sequence in an additional generation step to adapt the modules for the final output. However, Le et al. (2023) mainly targets traditional coding problems. For ML programs, this additional generation step is less effective (Table 1), due to the length of the programs. By comparison, our method automatically ensures the compatibility of generated modules and requires no extra generation step for combination.

Besides compositional reasoning, sequence generation methods for autoregressive LLMs also include iterative and prompting approaches. See Appendix A.1 for more related works of our method regarding LLMs.

## 2.2 Automated machine learning

AutoML methods typically involve an optimization algorithm and a search space (i.e., a set of candidate solutions) (Zoph & Le, 2016; Liu et al., 2018). Öztürk et al. (2022) explores the search space of pretrained deep learning models. Our autoML method adapts the strategies in Öztürk et al. (2022). In addition, we incorporate ZC proxies from Mellor et al. (2021) and Tanaka et al. (2020) in our method, which accelerates the optimization for deep learning tasks by predicting the performance of the models.

Very recently, several works have explored the application of LLMs for autoML, however, none of the works achieve automatic generation of the entire ML program. For a qualitative comparison, see Appendix A.2.

## 2.3 Software engineering

**Program synthesis** is a longstanding objective within the arctificial intelligence research community (Manna & Waldinger, 1971). Recently, there has been a surge of interest in ML for program synthesis, particularly with LLMs (Austin et al., 2021; Li et al., 2022). Conversely, program synthesis for ML is also an emerging topic and often involves autoML. Real et al. (2020) evolves programs for ML models. Saha et al. (2022); Cambronero & Rinard (2019) construct the programs for the modeling and part of data preparation for tasks about tabular data. Leveraging LLMs, our research expands the scope of program synthesis for ML by targeting the entire ML program across task modalities.

**Software testing** techniques (Myers et al., 2004) can assist program synthesis in evaluating the programs. Software testing for ML programs is a challenging topic (Braiek & Khomh, 2020). Pei et al. (2017) and Tian et al. (2018) address the challenge for computer vision tasks. Recently, Chen et al. (2022); Shinn et al. (2023) employ LLMs to automatically generate tests for traditional coding problems. Based on LLMs, the testing technique in our study is designed for implementing the Contextual Modular Generation framework. Compared to existing works, our technique is uniquely applicable for testing ML programs at a modular level and across task modalities (traditional algorithms, CV, and NLP).

# 3 Generation of Lengthy and Complex Sequences

## 3.1 Decomposition and compatibility

Compared to typical sequence generation tasks, code generation demands a complex output sequence. For example, generating 10 lines of code to meet a certain requirement is often considerably more challenging than an open-ended short writing of similar length. In addition, for an ML program, certain components—such as data preparation, modeling, and post-processing—require task-specific solutions. The components constitute a lengthy concatenated sequence, which can be many times longer than those found in traditional coding problems.

In this section, we introduce a sequence generation framework, *Contextual Modular Generation*, designed to scale effectively with the length of highly complex sequences (Figure 6 and Algorithm 1). The idea is straightforward: the required program is decomposed into distinct components—modules, which are then generated one at a time. As each module is generated, it undergoes an evaluation procedure (e.g., unit tests) to ensure it integrates seamlessly with the other modules—considering the context. If a module fails this evaluation, the generation process is iterated, using feedback from the evaluation procedure to refine the output.

This framework is a form of compositional reasoning (Andreas et al., 2016), which solves challenging problems by breaking them down into smaller, manageable components or modules. The framework also mirrors the intuitive practices of many *human programmers who break down a large program into manageable chunks, verifying the functionality and compatibility of each segment during its creation*. While the framework is generic for any sequence generation tasks under the availability of sequence evaluation methods defined in Definition 3.1, the framework is particularly effective in generating ML programs.

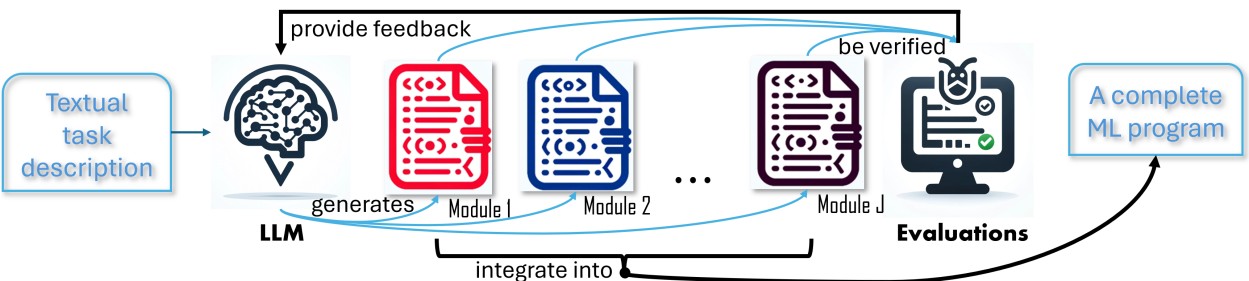

Figure 6: Contextual Modular Generation for ML tasks. First, the task description is given to the LLM. Then the LLM generates each module (i.e., the code for a step in ML) sequentially. Each module is generated in an iterative process, which includes the generation of code by the LLM, followed by verification via the evaluation methods (unit tests in Section 4), and incorporating feedback from these evaluations back into the LLM. In the end, the generated modules are automatically integrated into the ML program.

## 3.2 Formal definition

Consider the scenario where a binary evaluation method, denoted as $\mathbf{E}$, is available to test whether a generated sequence meets a specified requirement (e.g., absence of detectable errors and presence of desired functionality). For a given task, there exists a set of sequences that fulfill this requirement, represented as $\mathbf{\Omega} = (\omega_1, \omega_2, \ldots, \omega_C)$. Let $\mathbf{E}(Y)$ denote that the sequence $Y$ has been assessed by $\mathbf{E}$ and meets the specification. To establish a fair comparison framework applicable to all sequence generation methods (see Section 2.1 and Appendix A.1), we focus on the task of repeatedly generating the target sequence until a satisfactory sequence is produced. This task involves an LLM with state $\lambda$ and a set of input sequences $X_i$. In the $i_{th}$ iteration, the LLM generates an output sequence $Y_i$ using $X_i$. The process continues until the LLM produces a sequence $Y$, such that $Y$ can be any sequence or any concatenation of sequences generated in the current or previous iterations that meet the user's requirements as verified by $\mathbf{E}$.

Algorithm 1 defines our sequence generation framework (Figure 6). As a form of compositional reasoning, Algorithm 1 first divides the entire program into $J$ modules. Then for each module $j$, the algorithm repeatedly generates the code until success (lines 4 to 11). In the $i_{th}$ iteration, the LLM $f_\lambda$ first generates the code $Y^j$ using the instruction $X_i^j$ (line 8). Then, the evaluation method for the module $j$, $\mathbf{E}^j$, examines $Y_i^j$ and produce the feedback $R_i$ (line 9). In line 6, previous instructions $X_{<i}^j$, generated code $Y_{<i}^j$, and feedback $R_{<i}$ are assembled as the instruction $X_i^j$ for the current iteration.

A key premise of the framework is the availability of a set of evaluation methods as defined in Definition 3.1. These methods verify each module's individual functionality and compatibility with other modules.

---

**Algorithm 1** Contextual Modular Generation

---

1: **Input:** An autoregressive LLM $f_\lambda$, number of modules $J$, initial instructions for generating each module $\{X_0^1, X_0^2, \ldots, X_0^J\}$, contextual modular evaluations for each module $\{\mathbf{E}^1, \mathbf{E}^2, \ldots, \mathbf{E}^J\}$, and instruction constructors for each module $\{\Phi^1, \Phi^2, \ldots, \Phi^J\}$
2: **for** $j = 1$ **to** $J$ **do**
3:      $i = 1$
4:      **repeat**
5:          **if** $i > 1$ **then**
6:              $X_i^j = \Phi^j(X_{<i}^j, Y_{<i}^j, R_{<i})$
7:          **end if**
8:          $Y_i^j = f_\lambda(X_i^j)$
9:          $R_i = \mathbf{E}^j(Y_i^j)$
10:         $i = i + 1$
11:      **until** $R_i = True$
12:      $Y^j = Y_i^j$
13: **end for**
14: **Output:** $Y^1, Y^2, \ldots, Y^J$

---

**Definition 3.1.** For a task with desired sequences $\mathbf{\Omega}$, suppose the output sequences in one decomposition are $\{Y^1, Y^2, \ldots, Y^J\}$. For a set of binary evaluation methods $\{\mathbf{E}^1, \mathbf{E}^2, \ldots, \mathbf{E}^J\}$. We call the set of binary evaluation methods as *contextual modular evaluations* for the task, if that every $Y^j$ in $\{Y^1, Y^2, \ldots, Y^J\}$ is verified as valid by the corresponding $\mathbf{E}^j$ in $\{\mathbf{E}^1, \mathbf{E}^2, \ldots, \mathbf{E}^J\}$ guarantees that $\{Y^1, Y^2, \ldots, Y^J\}$ form a valid prgram (if $((\forall j, \mathbf{E}^j(Y^j)) \to \{Y^1, Y^2, \ldots, Y^J\} \in \mathbf{\Omega})$).

In other words, if the generated modules are verified by the contextual modular evaluations, then the modules are compatible with each other to form a desired program without any modification. Contextual modular evaluations eliminate the need for extra computation for adapting the generated modules into a sequence $Y$ such that $Y \in \mathbf{\Omega}$. In our study, the contextual modular evaluations verify individual components in the ML program to ensure consistency in data flow based on a novel testing technique (Section 4).

### 3.3 Theoretical analysis

Next, we theoretically analyze the scalability of Algorithm 1 over the length of complex output sequences. First, we define several metrics for comparing sequence generation methods. There are multiple ways to define the output length of a sequence generation problem (Anil et al., 2022). In our study, we define the *output length* of the task as the length of the longest valid output sequence: $\Gamma = max\{|Y||Y \in \mathbf{\Omega}\}$. Under the comparison scheme defined in Section 3.2, suppose the process of repeated generation terminates and the total number of iterations is $G(\Gamma)$ (We count $G(\Gamma)$ as $\infty$ if the iteration never terminates). Then the expectation of generations per success $\mathbb{E}_{x \sim P}[G(\Gamma)]$ is a measurement of the generation efficiency.

For a sequence $a$ and an integer $n$, let $\Theta_\Omega(a, n)$ denote the set of sequences $\{s | \exists s_r \wedge (\exists m \ (1 \le m \le \Gamma), |s| = \beta) \wedge |(a, s, s_r)| = m \wedge (a, s, s_r) \in \mathbf{\Omega}\}$, where $s_r$ is an arbitrary sequence (including the empty sequence) and $|a| + n \le \Gamma$. Intuitively, in a sequence generation process, $\Theta_\Omega(a, n)$ represents all valid sub-sequences from the $(|a| + 1)_{th}$ position to the $(|a| + n)_{th}$ position in the output sequence, given that the previously generated part in the output sequence is $a$. If $n = 1$, then $\Theta_\Omega(a, n)$ represents all valid tokens at the $(|a| + 1)_{th}$ position. For example, in Figure 7, let $t_i$ denotes the $i_{th}$ candidate token, then $\Theta_\Omega((t_2, t_4, t_6), 1) = \{t_1, t_3\}$.

**Definition 3.2.** For a task with desired sequences $\mathbf{\Omega}$, LLM $\lambda$, and input sequence X, let $\mathbf{p}_X^{\Omega, \lambda}(i)$ denote:

$$\mathbf{p}_X^{\Omega, \lambda}(i) = p_\lambda \left( y_i \in \Theta_\Omega(y_{<i}, 1) \cup \zeta(i) \ \middle| \ (y_{<i} \in \Theta_\Omega(\langle\rangle, i-1) \cup \eta(i-1)) \cap X \right)$$

where $\langle\rangle$ represents the empty sequence, $\eta(i-1)$ represents the event that the generation process terminates before the $(i-1)_{th}$ position in the output sequence, $\zeta(i)$ represents the union of $\eta(i)$ and that the generated sequence $\in \mathbf{\Omega}$, $y_i = \langle\rangle$ if $\eta(i)$, and $y_{<i} = \langle\rangle$ if $\eta(i-1)$.

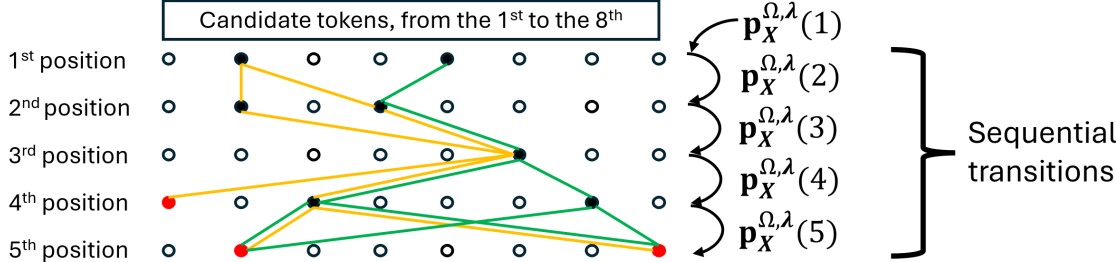

Figure 7: Sequential transitions in generating a sequence. The diagram provides a simplified illustration of a task in which the output sequence length, $\Gamma$, is 5, and the vocabulary size (i.e., number of choices for each token) is 8. Each solid dot represents a candidate token that could be included in a desired sequence at the specified position. Each red dot signifies a valid end-of-sequence (EOS) token, marking the conclusion of a sequence. The yellow and green lines each represent a tree structure, spanning all possible paths from a valid starting token to any EOS token, to construct a desired sequence. The cumulative probability of adhering to any path in any of the tree structures could decrease exponentially with each additional token generated.

We define *token-level complexity* at $\epsilon$ ($0 < \epsilon \leq 1$) for the task as:

$$\xi_X^{\Omega,\lambda}(\epsilon) = \left| \left\{ \mathbf{p}_X^{\Omega,\lambda}(i) \mid 1 \leq i \leq \Gamma \wedge \mathbf{p}_X^{\Omega,\lambda}(i) < \epsilon \right\} \right|$$

*Remark* 3.3. $\mathbf{p}_X^{\Omega,\lambda}(i)$ represents the probability that the generation step is correct for the $i_{th}$ position of the output sequence or a correct sequence with a length shorter than $i$ is generated. With this formulation, the generation process can be modeled as a Markov process with sequential transitions (successful token-level generation steps) and an absorbing state (a sequence with an incorrectly generated token) (Figure 7). $\xi_X^{\Omega,\lambda}(\epsilon)$ represents the number of tokens that are difficult to generate correctly (with a success rate less than $\epsilon$) and $0 \leq \xi_X^{\Omega,\lambda}(\epsilon) \leq \Gamma$ for any $\epsilon$.

**Assumption 3.4.** For the repeated generation task defined above, in every iteration, the token-level complexity at 1 is $\Gamma$.

**Theorem 3.5.** *For any sequence generation process, if there exists a step in the process that generates the entire output sequence and Assumption 3.4 holds for the step, then the best case complexity of $\mathbb{E}_{x \sim P}[G(\Gamma)]$ of the process is $\mathcal{O}(exp(\Gamma))$.*

*Remark* 3.6. Assumption 3.4 posits that errors may occur for any token of the output sequence, and the validity of this assumption is contingent upon $\mathbf{\Omega}$, $\lambda$, and $X$. Theorem 3.5 suggests that for traditional sequence generation approaches based on autoregressive LLMs, if the task exceeds a certain complexity level, the generation difficulty escalates exponentially over the length of the problem. If the value of $\xi_X^{\Omega,\lambda}(\epsilon)$ is ascertainable or estimable for a given $\epsilon$, regardless of the applicability of Assumption 3.4, deriving a more specific lower bound of $\mathbb{E}_{x \sim P}[G(\Gamma)]$ might be possible (Appendix B).

**Assumption 3.7.** For the repeated generation task defined above, in every iteration, for any $\epsilon > 0$, the token-level complexity at $\epsilon$ is $\Gamma$.

**Theorem 3.8.** *If Assumption 3.7 holds, then for Contextual Modular Generation as defined in Algorithm 1, the best case complexity of $\mathbb{E}_{x \sim P}[G(\Gamma)]$ is $\mathcal{O}(\Gamma)$.*

*Remark* 3.9. Assumption 3.7 imposes a more adverse condition compared to Assumption 3.4. Theorem 3.8 suggests that for Contextual Modular Generation, in the best case, the generation difficulty increases only linearly over the problem length, even for highly challenging tasks.

# 4 Unit Tests for Machine Learning

Contextual Modular Generation is conceptually intuitive, but its implementation demands meticulous design and non-trivial efforts. In this study, we utilize unit tests as contextual modular evaluations (as defined in

Definition 3.1) to ensure compatibility among the modules. The desiderata for the unit tests in our study are:

- As a basic role of unit tests, for each module, the tests should ensure the module is free from errors detectable by the code execution environment (e.g., Python interpreter) and confirm the presence of required functionalities, such as properties of the data processed by the module.

- Due to the high diversity in the code of the generated modules, the tests should be automatically generated. For instance, the required tensor shapes for the modules vary significantly depending on the ML tasks involved.

- Internal compatibility: In our method, for each module in the ML program, multiple code implementations of different candidate methods are generated to construct multiple candidate programs. (Figure 8 and Figure 3). Verified candidate codes of any module should be compatible with the candidate codes of other modules. This compatibility is crucial for the unit tests' role as contextual modular evaluations (as defined in Definition 3.1). Furthermore, for an ML program with $N$ modules, if $M$ candidate codes for each module are generated, this compatibility increases the ratio of generated program combinations over generation cost by a factor of up to $M^{N-1}$ (Figure 8).

- External compatibility: Verified modules should also be compatible with pre-written modules, including those implementing a consistent evaluation procedure, training procedure, testing procedure, and autoML methods (Figure 8).

These criteria require a balance between variation and constraint: each test targets a task-specific module while adhering to the principles of both internal and external compatibility. To address this unique challenge, we develop the testing technique *Constrained Generative Unit Testing*, as illustrated in Figure 8. Given that both the inputs and outputs of all modules in ML programs are data-centric, the technique predominantly ensures consistency in the structure and content of the data flow. A detailed formulation of the technique is available in Algorithm 3 of Appendix C.1.

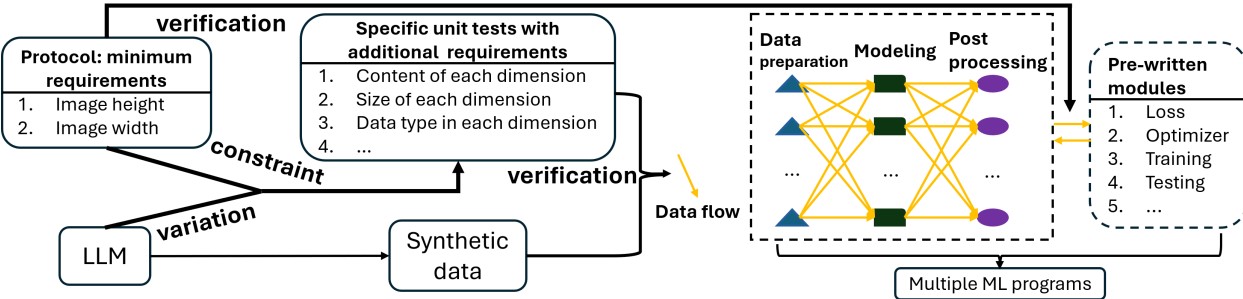

Figure 8: Constrained Generative Unit Testing for CV tasks. The yellow arrow represents the data flow between codes. The unit tests focus on verifying the structure and content of the data exchanged among the codes. For each module, codes for different candidate methods are generated. With the pre-written modules, each pathway through the network forms an ML program.

In the technique, the *progenitor testing protocol* is a suite of unit tests that serves as the foundational basis for the more specific unit tests generated by LLMs (Appendix C.1). This protocol may encompass one or several subsets of tests, each addressing a particular domain of tasks, such as NLP or CV. Each subset imposes minimum requirements that modules must meet for any task within the targeted domains. For instance, for typical CV tasks, the data processed by any modules should at least include one feature dimension for image height and another for image width. Besides addressing some of the requirements for internal compatibility, the primary purpose of the protocol is to ensure external compatibility—any generated modules are compatible with the set of pre-written modules for the domain.

The LLM specifies additional, more precise requirements for the given ML task. For example, besides image height and image width, the LLM specifies which other feature dimensions should be included and their appropriate sizes. These requirements extend the foundational constraints outlined in the protocol, ensuring that all specific instructions remain compliant with the protocal. The core mechanism of the technique lies in the interplay between the progenitor testing protocol and the LLM. The progenitor testing protocol establishes the minimal constraints shared across any task within the domain. Meanwhile, the LLM addresses the task-specific features required for the modules.

After the specific unit tests are constructed, the LLM generates multiple programs for producing synthetic data (Appendix C.1). This synthetic data serves as simulated input for the codes during unit testing. Every program for synthetic data is verified to adhere to the protocol. Hence, a code is considered valid upon successfully processing synthetic data from at least one program. Compared to actual data from the data preparation module, synthetic data can be deployed before the data preparation modules are generated and is independent of the specific data processing procedures.

## 5 Optimization of Machine Learning Programs

---
**Algorithm 2** Text-to-ML
---
1: **Input:** An autoregressive LLM $f_\lambda$ and textual description of an ML task $\mathbf{T}$
2: $\mathbb{H} = [\,]$ and $\mathbb{S} = \textit{Search Space Generation }(\mathbf{T}, f_\lambda)$
3: **for** $\theta$ in $\mathbb{S}$ **do**
4:      $\mathbf{C}_{\mathbf{M}}^\theta = \textit{Module Generation }(\mathbf{M}, \theta, f_\lambda)$ // $\mathbf{M}$ represents the module to be generated is for modeling
5: **end for**
6: **if** based on $\mathbb{S}$, the task is a deep learning task **then**
7:      $\mathbf{S} = \textit{ZC Proxies Evaluation }(\mathbb{S}, \{\mathbf{C}_\theta^m\})$
8: **end if**
9: **repeat**
10:      $\theta = \textit{Search Strategy }(\mathbb{S}, \mathbb{H})$
11:      **for** module $j$ in $\theta(j \neq \mathbf{M})$ **do**
12:          **if** no prior attempt exists for generating $\mathbf{C}_j^\theta$ **then**
13:              $\mathbf{C}_j^\theta = \textit{Module Generation }(j, \theta, f_\lambda)$
14:          **end if**
15:      **end for**
16:      **if** All syntactically valid modules for $\theta$ exist **then**
17:          Evaluate the ML program based on $\theta$, with evaluation result as $R_\theta$, then append $\theta$ and $R_\theta$ to $\mathbb{H}$
18:      **end if**
19: **until** resources are exhausted
20: **Output:** The best ML program based on $\mathbb{H}$
---

As Figure 8 illustrates, a network of codes is generated to form multiple candidate ML programs. AutoML then selects the high-performance program (i.e., pathway within the network), along with associated hyperparameter values. Leveraging the external compatibility (Section 4), autoML is implemented as pre-written modules.

Many autoML methods architect high-performance deep learning models from scratch—such as determining the number of convolutional layers and initiating training from zero. Such a process is prohibitively expensive for regular users under low resource constraints (Öztürk et al., 2022). By comparison, the search space in our study leverages the ability of LLMs to directly deploy pretrained models as candidate methods for modeling.

As a similar search space is explored in Öztürk et al. (2022), we employ a comparable set of hyperparameters as in Öztürk et al. (2022) (Appendix C.7). By contrast, the search space in our study includes more candidate pretrained models and both CV and NLP models. To manage the expanded scope, our autoML method incorporates four ZC proxies applicable with synthetic data for both CV and NLP (Appendix C.5), which significantly mitigates the cost associated with training and testing ML programs (Figure 4).

Algorithm 2 outlines our comprehensive methodology, from code generation to code optimization, as illustrated in Figure 3. In Algorithm 2, the LLM $f_\lambda$ initially creates the search space $\mathbb{S}$ with *Search Space Generation* (Appendix C.2), suggesting candidate algorithms for each module, such as candidate feature engineering techniques and ML models, and candidate hyperparameter values (line 2). Then for the ML model in each candidate solution of the search space $\theta$, the LLM generates the corresponding code for the model $\mathbf{C}_{\mathbf{M}}^\theta$ with *Module Generation* (Appendix C.4) (lines 3 to 5). *Module Generation* generates the code with the LLM based on Algorithm 1 and Algorithm 3.

Notably, with the synthetic data from Algorithm 3, for deep learning tasks, the modeling modules are evaluated with *ZC Proxies Evaluation* (Appendix C.6) before generating other modules, saving both generation and optimization costs (lines 6 to 8). *ZC Proxies Evaluation* removes a fraction of lower-performing candidate solutions from the search space with minimal cost.

Subsequently, Algorithm 2 iteratively examines the candidate solutions in the search space through trial-and-error. In each iteration, with the previous optimization history $\mathbb{H}$ and search space $\mathbb{S}$, *Search Strategy* (Appendix C.3), which is an optimization algorithm, selects a promising solution for the ML task (line 10). Then, for the selected solution, *Module Generation* generates all modules except the modeling module (lines 11 to 15). Once all valid modules for the solution are created, Algorithm 2 assembles the complete ML program, evaluates it using the designated numerical metrics in $\mathbf{T}$, and records the solution and its performance in the optimization history.

Ultimately, Algorithm 2 delivers the code for the ML task that exhibits the highest performance among all evaluated solutions. The code is an executable program that seamlessly integrates the modules.

# 6 Experiments

## 6.1 Setup

To explore the limits of program synthesis for ML, in experiments, we consider the synthesis of the entire ML program, given textual descriptions of the ML task. For an ML task, let the set of valid programs (as defined in Section 6.3) for the entire ML program of the task be the desired sequences $\mathbf{\Omega}$. Given the textual description $\mathbf{T}$, testing data $D^{tes}$, and evaluation metrics $\mathbf{E}$, then the objective is to automatically generate the program $\mathbf{C}_\theta$ that maximizes the evaluation score based on $\mathbf{E}$ and $D^{tes}$:

$$\arg\max_\theta \mathbf{E}\left(\mathbf{C}_\theta, D^{tes}\right) \quad \text{s.t. } \mathbf{C}_\theta \in \mathbf{\Omega}$$

In other words, at first, a precondition is that the generated programs are free of major syntax errors. Then among these programs, select the best program, by maximizing the performance score evaluated by the metrics. For other sequence generation methods, which are incompatible with autoML due to the lack of external compatibility (see Section 4), we only evaluate their efficiency in code generation (Section 6.2).

For our method, we further perform code optimization (Section 6.4). We generate each candidate program $\mathbf{C}_\theta$ in an iterative approach through LLMs and autoML. Given an LLM $f_\lambda$, generation method $\Psi : f_\lambda \times \mathbf{T} \times \theta \to \mathbf{C}_\theta^*$, autoML method $\Theta$, training data $D^{tr}$, and training procedure $\mathbf{H} : \mathbf{C}_\theta^* \times D^{tr} \times \theta \to \mathbf{C}_\theta$, in the $i_{th}$ iteration, our method generates $\mathbf{C}_i$:

$$\mathbf{C}_i = \mathbf{H}\left(\mathbf{C}_\theta^*, D^{tr}, \Theta(i)\right), \text{ where } \mathbf{C}_\theta^* = \Psi\left(f_\lambda, \mathbf{T}, \Theta(i)\right)$$

**Setup of datasets**   For the novel problem described above, we curate a suite of datasets in our experiments by carefully addressing the relevant factors. For each of the traditional algorithms, CV and NLP, the datasets include 2 widely-recognized datasets for traditional studies (Appendix D.1) and 2 datasets from the Kaggle competitions (Appendix D.2). The widely recognized datasets can be less biased; however, the training corpus of the LLMs may encompass knowledge about these datasets. For the Kaggle datasets, we conduct a thorough review of all medal-awarding competitions held between October 2021 and July 2023 (Appendix D.2), considering the knowledge cutoff dates of the LLMs.

**Setup of methods** We use GPT-4-0613, GPT-3.5-turbo-0613 (Achiam et al., 2023), and PaLM 2 (Anil et al., 2023) as LLMs $f_\lambda$. In Algorithm 1, we incorporate the self-reflection mechanism in Shinn et al. (2023) in $\{\Phi^j\}$ (see Appendix A.1). We employ both random search and the hyperparameter optimization method BOHB (Falkner et al., 2018) as the algorithms for code optimization (see Appendix C.3). For ZC proxies, we utilize *flops*, *params*, *naswot*, and *synflow* (Appendix C.5). *flops* and *params* are two simple yet powerful ZC proxies (Krishnakumar et al., 2022). While most ZC proxies are designed for CV tasks, Zhou et al. (2022b) and Javaheripi et al. (2022) suggest the suitability of *naswot* and *synflow* for NLP tasks. Throughout the experiments, for the same dataset, we provide the same task description **T** for each generation method (Appendix E).

## 6.2 Generation of valid programs

As outlined in the task of repeated generations (Section 3), we compare each sequence generation method across the datasets by repeatedly attempting to generate valid ML programs. This process is capped at a maximum of 100 attempts. If a method generates a valid program within these 100 attempts, the method is reset and tasked with generating another valid program until all 100 attempts are exhausted. For all methods, particularly Contextual Modular Generation, each sequence generation process (each time line 8 in Algorithm 1 is executed) counts as an attempt.

In Table 1, among the techniques, Reflexion and Contextual Modular Generation incorporate a self-reflection mechanism, as outlined by Shinn (2023). This mechanism retains memories of previous trials, including feedback from the testing results of earlier iterations (see Figure 3 and Algorithm 1). Such feedback is used as an input sequence to provide insights for the generation process. However, we observe that these two methods often fall into repetitive cycles involving the same code-errors-feedback loop. To address this issue, we force a memory reset if the method fails to produce a valid program after 10 attempts, thus mitigating the effects of poor initiations. For CoT, we set the number of input-output examples as 3 for GPT-4 and GPT-3.5 (Table 6) and 2 for PaLM 2 (Table 7).

Table 1: Mean number of attempts per valid program with GPT-4. (1) CoT, Zero-shot, and Reflexion refer to Wei et al. (2022), Kojima et al. (2022), and Shinn et al. (2023) respectively. (2) Self-Revision: the generation based on the replacement of the contextual modular evaluations in Algorithm 1 with the self-revision mechanism in Le et al. (2023). (3) for Contextual Modular Generation, w/o tests: Algorithm 1 by removing the unit tests (Figure 8 and Algorithm 3), w/o reflection: Algorithm 1 by removing the self-reflection mechanism in $\{\Phi^j\}$, standard: Algorithm 1. (4) the datasets are divided into three groups in the order of tasks for traditional algorithms, CV, and NLP. Details of each dataset are available in Appendix D. (5) -: the method is unable to generate any valid program for the dataset within 100 attempts.

| Datasets | CoT | Zero-shot | Reflexion | Self-Revision | Contextual Modular Generation (ours) | | |
|---|---|---|---|---|---|---|---|
| | | | | | w/o tests | w/o reflection | standard |
| Boston | $10.2 \pm 2.2$ | - | $3.7 \pm 0.8$ | $4.0 \pm 0.6$ | $4.1 \pm 1.0$ | $3.9 \pm 0.7$ | $\mathbf{3.4 \pm 0.6}$ |
| Iris | $\mathbf{2.1 \pm 0.5}$ | $3.5 \pm 0.6$ | $6.1 \pm 0.8$ | $3.7 \pm 1.1$ | $3.6 \pm 0.7$ | $3.4 \pm 0.6$ | $3.2 \pm 0.4$ |
| *Age* | $21.6 \pm 7.6$ | - | $13.0 \pm 4.0$ | $8.8 \pm 2.9$ | - | $9.1 \pm 1.0$ | $\mathbf{7.1 \pm 0.9}$ |
| *Default* | $34.3 \pm 9.0$ | - | $14.7 \pm 5.6$ | $13.1 \pm 1.6$ | - | $\mathbf{10.6 \pm 3.2}$ | $11.6 \pm 2.1$ |
| CIFAR-10 | $4.5 \pm 1.1$ | $6.3 \pm 1.8$ | $\mathbf{4.3 \pm 0.5}$ | $13.6 \pm 1.6$ | - | $6.9 \pm 1.6$ | $6.7 \pm 1.5$ |
| CIFAR-100 | $7.2 \pm 1.4$ | $6.6 \pm 1.2$ | $13.3 \pm 3.9$ | $20.4 \pm 3.9$ | - | $8.3 \pm 3.6$ | $\mathbf{5.6 \pm 0.9}$ |
| *Whale* | - | - | $30.5 \pm 13.4$ | - | - | $\mathbf{9.7 \pm 0.8}$ | $11.0 \pm 1.2$ |
| *Strip* | - | - | - | - | - | $12.5 \pm 2.4$ | $\mathbf{9.9 \pm 1.1}$ |
| IMDb Reviews | $6.0 \pm 1.5$ | $4.4 \pm 1.1$ | $6.2 \pm 1.4$ | $10.5 \pm 1.6$ | - | $11.1 \pm 1.5$ | $\mathbf{3.5 \pm 0.5}$ |
| AG News | $12.0 \pm 1.1$ | $15.1 \pm 3.2$ | $9.6 \pm 1.4$ | $17.3 \pm 2.9$ | - | $7.4 \pm 1.5$ | $\mathbf{4.2 \pm 1.2}$ |
| *Exam* | - | - | - | - | - | $12.1 \pm 3.1$ | $\mathbf{10.7 \pm 2.6}$ |
| *Learning* | - | - | $20.0 \pm 2.8$ | - | - | $\mathbf{12.6 \pm 1.6}$ | $12.8 \pm 1.6$ |

In Table 1, we report the average number of attempts required to generate one valid ML program. Overall, our method outperforms other methods in 10 out of 12 of the datasets. For ML tasks with longer program lengths,

especially the deep learning tasks and those from Kaggle, Contextual Modular Generation significantly outperforms other methods. In particular, prompting methods (CoT and Zero-Shot) are generally unable to address these tasks.

The performance gap between Contextual Modular Generation w/o reflection and Contextual Modular Generation standard is narrow, and in some instances, the former even shows superior results. This could be attributed to the possibility that the lengthy memory involving previously generated ML programs is overwhelming and counterproductive. When we remove unit tests (Algorithm 3) from the process (Contextual Modular Generation w/o tests), there is a noticeable increase in the generation difficulty across datasets (Contextual Modular Generation w/o tests in Table 1), which suggests that the testing technique (Figure 8 and Algorithm 3) is necessary for Contextual Modular Generation in most situations.

We also conduct experiments with GPT-3.5 and PaLM 2 (Appendix F). For both LLMs, Text-to-ML outperforms other methods in 11 out of the 12 datasets. Overall, GPT-4 demonstrates the highest success rate among the LLMs.

For the code generation process, we analyze the mean number of iterations required to correct the errors (Table 8 in Appendix F). In the experiments, for code that fails unit tests, we calculate the average number of iterations needed to pass the tests again. We only count corrections before the forced memory reset. Overall, (1) data preparation modules are generally the most difficult to correct, averaging 5.43 iterations; (2) for GPT-4, with its larger input token limit, the self-reflection mechanism shows the greatest benefit; (3) for LLMs with smaller input token limits (GPT-3.5 and PaLM 2), the memory from self-reflection can become unmanageable when a single previous iteration exceeds the input token limit, resulting in automatic memory truncation.

### 6.3 Unit tests and false positives

In a strict sense, errors in ML programs are nearly impossible to eliminate compared to traditional coding problems. Traditionally, software systems are constructed deductively by explicitly programming the rules that govern system behavior. In contrast, ML rules are inferred inductively from the data (Braiek & Khomh, 2020). For example, the program may be error-free for one set of data points but buggy for another (Kim et al., 2019; Braiek & Khomh, 2020; Wang et al., 2021b). Moreover, many ML programs rely on high-level libraries like PyTorch and TensorFlow, and third-party libraries such as the Intel Math Kernel Library, which often lack comprehensive specifications or even detailed source code documentation (Braiek & Khomh, 2020).

Nevertheless, as in the practices of many human programmers, ML programs can still be *free of major errors and function effectively in most daily scenarios*. In our study, we define valid programs as those that satisfy two criteria: (1) the program executes over the datasets in the task without interruptions from errors detectable by the execution environment (the Python interpreter), and (2) for programs that meet the first requirement, we manually ensure that the programs form a complete program for the specified ML task. Specifically, we manually check whether the generated program includes a complete ML pipeline, including data preparation, modeling, and post processing, as specified by the LLM in *Search Space Generation* in Algorithm 3 (see Appendix C.4).

Similar to traditional coding problems (Li et al., 2022), an ML program can be a *false positive*, appearing valid in unit tests but actually failing to meet the criteria mentioned above. In our study, each program is divided into modules, each verified by unit tests before assembly. However, modules verified as valid individually may not necessarily remain valid when combined. Therefore, the entire assembled program undergoes an additional simple test by the execution method to automatically verify the first criterion.

In the experiment in Section 6.2, for our method (Contextual Modular Generation - standard Table 1), we examine the false positive (FP) rates of the generated programs both before and after this additional testing step.

As Table 2 shows, the false positive rates for deep learning tasks are significantly higher than those of traditional algorithms, but they are still within a reasonable range. The false positive rates of the programs after meeting the first criterion are noticeably low, which suggests the reliability of ML programs that can be created automatically (by employing the execution step as an additional testing step).

Table 2: Unit tests efficiency and false positive rates. (1) Traditional: tasks based on traditional ML algorithms such as random forest and SVM. (2) FP rate w/o execution: false positive rates of the programs post-verification but prior to testing by the execution environment (3) FP rate w/o execution: false positive rates of the programs following both verification and testing by the execution environment.

| Modality | ML programs verified by unit tests | FP rate w/o execution | FP rate w execution |
|---|---|---|---|
| Traditional | 86 | 7% | 0% |
| CV | 79 | 34% | 2% |
| NLP | 99 | 29% | 0% |

Notably, for the experiments in Section 6.2, we do not insist on the compatibility of generated modules with any pre-written module, which is more lenient towards the other methods, particularly those without testing, such as CoT and Zero-Shot. Achieving such external compatibility (Section 4), is infeasible for methods that do not include testing or rely on traditional testing techniques. However, in our method, the generated modules are also tested for compatibility, which is crucial for performing consistent training and testing procedures and for a uniform evaluation of these modules.

### 6.4 Improvements from optimization

Our method ensures that generated programs are compatible with a consistent evaluation procedure. Leveraging this unique advantage, Text-to-ML automatically searches the programs generated by Contextual Modular Generation against metrics specified in the textual task descriptions. For each dataset, with GPT-4, we generate codes of up to 20 candidate algorithms, each for data preparation and modeling and at least 1 code for post processing, forming up to 400 module combinations due to the internal connectivity (Section 4) provided by the testing technique (the network in Figure 8).

From these modules, we first randomly sample and evaluate 200 configurations of module combinations (pathways in the network in Figure 8) and their hyperparameters. We employ two optimization strategies: random search and BOHB (Falkner et al., 2018) (Appendix C.3). For both strategies, we examine the identified program at the cost of 25 and 100 full evaluations (i.e., the cost equivalent to the cost of fully training and testing 25 and 100 models). In the search, ZC proxies and BOHB prematurely terminate the evaluations for less promising configurations (Appendix C.3). Then, we rank the best program identified during the search among the 200 sampled configurations.

For deep learning tasks, we train each model on 1 of 4 NVIDIA A100 GPUs. For sampling the configurations and random search, we train each model until it meets the early stopping criteria with a *patience* of 3 and a *minimum delta* of 0.0 (Prechelt, 2002).

The programs generated by traditional sequence generation methods are incompatible with autoML; however, these methods do allow for the program to be specified by LLMs. As a simple baseline, we also evaluate the one ML program chosen by the LLM without any further optimization (No Search in Table 3). We utilize the prompt *select the best configuration for the machine learning task.*

Table 3 shows random search already yields significant improvements over the baseline, particularly with cost@25. In addition, ZC proxies effectively increase the search efficiency for both CV and NLP, improving the ranks of programs from the search with cost@25 by $31 \pm 12\%$ and the search with cost@100 by $9 \pm 3\%$.

## 7 Discussion

### 7.1 Synergy between LLMs and autoML

A major goal of autoML is to democratize ML by reducing the amount of effort required by human experts (Thornton et al., 2013; Wang et al., 2021a; Xin et al., 2021; Van der Blom et al., 2021). AutoML has achieved substantial progress in recent years on an algorithm level (Zoph & Le, 2016; Liu et al., 2018). However, in practice, autoML methods are gradually becoming a tool for experts to potentially boost the performance

Table 3: Mean rank of the identified ML program in 10 trials. (1) Traditional: tasks based on traditional ML algorithms such as random forest and SVM. (2)Rows with white and grey backgrounds represent search results with cost@25 and cost@100 respectively.

| Modality | No Search | Search only | | Search + ZC proxies | |
|---|---|---|---|---|---|
| | | Random search | BOHB | Random search | BOHB |
| Traditional | 97.4 ± 41.0 | **10.8 ± 7.2** | 10.8 ± 9.3 | N/A | N/A |
| | | 5.0 ± 2.8 | **4.2 ± 1.9** | N/A | N/A |
| CV | 75.2 ± 34.9 | 10.2 ± 4.7 | 8.9 ± 5.8 | **6.2 ± 4.2** | 7.5 ± 5.4 |
| | | 4.3 ± 1.7 | 4.0 ±1.1 | 3.8 ± 0.8 | **3.6 ± 0.7** |
| NLP | 93.8 ± 31.4 | 8.5 ± 6.6 | 10.7 ± 5.4 | 6.6 ± 4.7 | **5.8 ± 2.4** |
| | | 4.7 ± 2.1 | 4.0 ±1.5 | 4.2 ± 0.7 | **3.8 ± 1.0** |

of the workflow (Tornede et al., 2023; Van der Blom et al., 2021), due to the following three issues: (1) most autoML methods only select the solutions at the algorithm level. However, the programming required to implement the algorithms can demand substantial human effort and expertise. (2) for deep learning, many autoML methods select model architectures, such as the number of convolutional layers, from scratch and train the models anew, which is prohibitively expensive for regular users in practical scenarios under low resource constraints (Öztürk et al., 2022). (3) many autoML methods do not offer a comprehensive and unified solution applicable across task modalities (e.g., traditional algorithms, CV and NLP).

In our study, autoML numerically evaluates and selects ML programs generated by LLMs. Conversely, LLMs offer a solution to all three issues of autoML. As demonstrated in Figures 1 and 9 and discussed in Section 6, LLMs can generate the programs for the entire ML workflow across task modalities, while directly utilizing pretrained models for deep learning. Without the code generation from LLMs, there would be no code for autoML to select and optimize in the first place. Our approach not only generates this code but also ensures that the code is natively compatible with optimization algorithms.

The objectives of autoML and program synthesis for ML are closely aligned, with significant overlap in their purposes and methods (see Section 2.3). Nonetheless, strictly speaking, the goal of our study is on program synthesis for ML, rather than autoML per se. This distinction arises because most autoML methods do not address ML tasks at the code level. In this context, autoML is employed as a step in our method (Figure 3).

*For the goal of synthesizing ML programs, LLMs synergize with autoML: LLMs bridge the gap between algorithm-centered autoML and autoML in practice, and autoML addresses the complex numerical evaluations of the programs generated by LLMs.* In this synergy, LLMs contribute extensive coding knowledge embedded in their weights, facilitating code generation, while autoML brings robust mathematical rigor from numerical optimization, enabling code optimization.

## 7.2 Sequence generation and autonomous agent

From a sequence generation perspective, our work explores the resolution of ML problems as a multi-step sequence to sequence (seq2seq) task (Sutskever et al., 2014). The input sequence is the textual task description. The output sequence is the corresponding optimized program for the entire ML program. Intermediary steps include sub-seq2seq tasks based on LLMs and optimization via autoML.

Our method, *Text-to-ML*, seamlessly integrates all intermediary steps into a single fully autonomous process (Figure 3 and Algorithm 2). Future research may explore such multi-step seq2seq tasks as autonomous agents. Another interesting future exploration involves incorporating text-to-text or image-to-text methods to enhance task description generation, further automating the process.

### 7.3 Limitations

While the complexity of Contextual Modular Generation could increase only linearly over the problem length (Theorem 3.8), implementing this framework demands considerable effort (Appendix C.1). The practical significance of the theoretical linear complexity should be more extensively tested for other types of sequence generation tasks. Furthermore, our approach still struggles with highly challenging ML tasks, particularly those that involve highly complex forms of input and output data (Appendix D). Whether addressing these tasks requires a more efficient implementation of the framework or a transition to a different framework remains an open problem.

In particular, challenging tasks on the Kaggle platform are typically addressed through the collective efforts of thousands of participants. The participants can freely exchange ideas and solutions, greatly enhancing the efficiency of developing high-performance solutions. Additionally, the highest-performing publicly shared solution often becomes the default baseline, as many participants replicate it in their solutions. These practices make a meaningful comparison between the solutions from human participants and automated solutions infeasible. In practice, most ML tasks are tackled by only one or a few human programmers.

Regardless, our work is not intended to outperform the collaborative efforts of thousands of competition participants with automatic solutions. Instead, our focus is on reducing the amount of effort and knowledge required for writing ML programs in everyday scenarios. Qualitatively, our method achieves an unprecedented degree of automation for ML tasks at the code level. Additionally, our work demonstrates significant performance improvements in the generated ML programs from the automatic optimization process.

To target performance enhancement in a competitive environment, a future direction could involve significantly expanding the number of candidate solutions. For example, in our method, a modeling module might consist of a pretrained model and a custom architecture designated by the LLMs (Appendix C.2, Figure 1 and Figure 9). Future research could extend Text-to-ML with additional autoML techniques, such as predictor-based methods (Dudziak et al., 2020) and weight-sharing methods (Pham et al., 2018), to search for the custom architecture. Whether and to what extent such a pursuit can coexist with applications for regular users under low resource constraints is a crucial consideration. Additionally, establishing a systematic benchmark to compare automated solutions with human-generated ones is another significant challenge in this direction.

## 8 Conclusion

This study extends the scope of program synthesis for ML. By combining LLMs and autoML, we achieve the automatic generation and optimization of the complete ML programs over 12 tasks across traditional algorithms, CV, and NLP. Experimentally, over 10 out of the 12 tasks, our generation framework Contextual Modular Generation outperforms existing methods, particularly for deep learning tasks and tasks outside the training corpus of the LLMs. Theoretically, Contextual Modular Generation offers a linear best-case complexity class for generation difficulty over sequence length.

The testing technique Constrained Generative Unit Testing is instrumental for the implementation of the framework. A notable advantage of our approach is the seamless compatibility between generated and pre-written modules, allowing for consistent numerical evaluation and native integration with optimization algorithms. This compatibility enables autoML-driven numerical optimization to substantially improve program performance. In addition, we demonstrate the effectiveness of ZC proxies for pretrained models in both CV and NLP.

### Broader Impact Statement

From the perspective of autoML, our approach aims to enhance the automation of the ML workflow, especially for everyday scenarios with limited resources. Text-to-ML is a potential tool for democratizing ML by significantly reducing the required efforts, expertise, and computational resources. Users simply provide a textual description of the task, and from there, the process is fully automatic, making ML in practice accessible to a broader audience.

However, the broader impacts of Text-to-ML must be carefully considered to ensure its responsible and ethical use. Our commitment to responsible AI is paramount, and we strongly urge the responsible use of automated tools for ML, including Text-to-ML.

**Accessible Machine Learning: Benefits and Risks**

By enabling a broader range of individuals and organizations to leverage ML, the Text-to-ML framework can foster innovation, accelerate scientific discovery, and enhance decision-making across various fields. However, this increased accessibility also introduces potential risks, such as misuse by individuals with malicious intent or those lacking a proper understanding of ML principles. We list the guidelines for responsible use of Text-to-ML as below:

- Ethical Education: Users should complete a general ethics course in ML/AI before accessing Text-to-ML. This course would cover topics including fairness, bias, and risks in ML, and provide examples of both responsible and irresponsible uses of ML technology.

- Bias and Toxicity in Data: Before applying Text-to-ML to a dataset, we recommend thoroughly checking for common biases such as selection bias, label bias, and sampling bias to ensure the dataset represents diverse populations, avoids subjective influences, and uses appropriate sampling methods (Mehrabi et al., 2021). Additionally, it is crucial to filter out toxicity in the data, including hate speech, offensive content, and misinformation.

- Data Privacy: Users of Text-to-ML must adhere to relevant data privacy laws and regulations, such as the General Data Protection Regulation (GDPR) (Voigt & Von dem Bussche, 2017) in the European Union and similar laws in other jurisdictions. For instance, using Text-to-ML on personal data without consent or scraping data from unauthorized sources would be strictly prohibited.

- Application Restrictions: The use of Text-to-ML for irresponsible or harmful applications should be explicitly prohibited. Examples include generating misleading or harmful content and developing surveillance systems that infringe on individual privacy rights.

- Impact Assessment: Users should conduct an impact assessment for their ML programs using Text-to-ML, outlining the potential societal and ethical implications. For example, a project aimed at predicting criminal behavior would need to consider the risks of reinforcing biases and stigmatizing individuals or communities.

**Transparency and Accountability in the Automation of Machine Learning**

Automating the synthesis of ML programs streamlines the creation of ML models, but it can also obscure the underlying decision-making process. This lack of transparency is particularly concerning in critical fields such as healthcare, finance, criminal justice, job hiring, and college admissions. For instance, in finance, transparency is necessary to ensure fair lending practices and avoid biases in credit scoring. In college admissions, understanding the rationale behind acceptance or rejection decisions is vital for maintaining fairness and avoiding biases that could disadvantage certain groups of applicants. We outline our measures and guidelines to increase transparency and accountability for Text-to-ML as follows:

- Logging the Automatic Process: The implementation of Text-to-ML in our study features a comprehensive logger that records the entire process of synthesizing the ML program. This includes decisions regarding the search space construction (Appendix C.2), rationales behind the suggestions of candidate solutions (Figure 9), successfully generated modules, and the performance of each module combination being evaluated. We strongly recommend that other implementations of the Text-to-ML framework also incorporate such a logger. This detailed logging information helps users trace outcomes back to specific decisions and increases accountability.

- Tracing the Prompts and Responses of LLMs: Text-to-ML involves a set of dynamically constructed prompts (Appendix E) in intermediate steps through the instruction constructors ($\{\Phi^1, \Phi^2, \ldots, \Phi^J\}$ in Algorithm 1). The implementation of Text-to-ML in our study transparently displays the input

and output sequences for each inquiry involving LLMs. This history of prompts and responses enables users to understand the decisions made by the LLMs in each step of the process, enhancing transparency and traceability.

- Interpretable Models: We advocate for the use of interpretable models and techniques that allow users to understand how specific outcomes are derived. This includes the incorporation of explainable AI (XAI) methods (Hoffman et al., 2018) within the Text-to-ML framework to provide insights into the decision-making process.

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

# A More Related Works

## A.1 Large Language Models

**Iterative methods** repeatedly generate the output sequences in order to improve the output sequence based on a sequence evaluation procedure. Let the initial input sequence be $X_0$, the sequence generation process be $f_\lambda$, and the evaluation procedure be $\mathbb{E}$. Then the output sequence in the $i_{th}$ iteration is $Y_i = f_\lambda(\Phi_i(Y_{<i}, \mathbb{E}(Y_{<i}), X_0))$, where $\Phi_i$ represents a mechanism that assembles the input sequence for the iteration by integrating the output sequences, their evaluations, and the initial input sequence from preceding iterations. Recent works along this line include Yao et al. (2022); Madaan et al. (2023); Cobbe et al. (2021); Shinn et al. (2023); Chen et al. (2023b). In a reinforcement learning context, each output sequence

represents an action (Yao et al., 2022; Shinn et al., 2023). The implementation of our method incorporates the self-reflection mechanism in Shinn et al. (2023). More specifically, $\Phi_i$ in our implementation assembles input sequence from code execution results and an explanation of the results.

**Prompting methods** improves the output sequence $Y$ by transforming the input sequence $X$. Let $\psi$ represents the prompt engineering process, then $p(Y|X) = \prod_{n=1}^{N} p_\lambda(y_n|y_{<n}, \psi(X))$. Notable prompting methods include in-context learning (Wei et al., 2022; Zhou et al., 2022a; Wang et al., 2022; Min et al., 2022; Yao et al., 2023; Li et al., 2023) and zero-shot prompting (Kojima et al., 2022). Notably, although some prompting methods break the problem into intermediate steps, such as Wei et al. (2022) and Li et al. (2023), these methods perform the steps in one sequence generation process, differing from our definition of compositional reasoning (see Section 2.1).

## A.2 Automated Machine learning

Table 4 compares our method with existing methods that employ LLMs for autoML. A detailed analysis of recent advancements in this domain can be found in Tornede et al. (2023). To the best of our knowledge, Text-to-ML is the first method that automatically generates the optimized program of the entire ML program across task modalities.

While recent autoML techniques also include predictor-based methods (Dudziak et al., 2020) and weight-sharing methods (Pham et al., 2018), these techniques are less suitable for the desiderata in our study: applications in low resource constraints scenarios and search space with diverse pretrained deep learning models.

Table 4: Comparison with existing methods in terms of qualitative features. (1) Code generation: whether the method automatically generates programs (2) Deep learning: whether the method automatically generates (part of) programs for deep learning tasks. (3) Data preparation: whether the method automatically chooses the strategy for data preparation (4) Modeling: whether the method automatically chooses the strategy for modeling (5) Optimization: whether the method optimizes the programs (components). * Although the method can automatically choose the strategy for modeling, the method does not automatically generate the code for modeling. ** The method uses the LLM as a pseudo-optimization model instead of a dedicated optimization algorithm (e.g., evolutionary computation and reinforcement learning.)

| Methods | Code generation | Deep learning | Data preparation | Modeling | Optimization |
|---|---|---|---|---|---|
| GPT-NAS (Yu et al., 2023) | | | | ✓* | ✓ |
| MLCopilot (Zhang et al., 2023a) | | | | ✓* | |
| GENIUS (Zheng et al., 2023) | | | | ✓* | ✓** |
| Evoprompting (Chen et al., 2023a) | ✓ | ✓ | | ✓ | ✓** |
| CAAFE (Hollmann et al., 2023) | ✓ | | ✓ | | |
| SapientML (Saha et al., 2022) | ✓ | | ✓ | ✓ | |
| AutoML-GPT (Zhang et al., 2023b) | ✓ | ✓ | ✓ | ✓ * | |
| HuggingGPT (Shen et al., 2023) | ✓ | ✓ | | ✓ | |
| **Text-to-ML** | ✓ | ✓ | ✓ | ✓ | ✓ |

# B Proofs of Theorems

We start by connecting the concept of token-level complexity (Definition 3.2) with the success rate of generation.

In Definition 3.2,

$$\mathbf{p}_X^{\Omega,\lambda}(i)$$

$$= p_\lambda\Big(y_i \in \Theta_\Omega(y_{<i}, 1) \cup \zeta(i) \;\Big|\; (y_{<i} \in \Theta_\Omega(\langle\rangle, i-1) \cup \eta(i-1)) \cap X\Big)$$

$$= \frac{p_\lambda((y_i \in \Theta_\Omega(y_{<i}, 1) \cup \zeta(i)) \cap ((y_{<i} \in \Theta_\Omega(\langle\rangle, i-1) \cup \eta(i-1)) \cap X))}{p_\lambda((y_{<i} \in \Theta_\Omega(\langle\rangle, i-1) \cup \eta(i-1)) \cap X)}$$

Since $(y_i \in \Theta_\Omega(y_{<i}, 1))$ and $\zeta(i)$ are mutually exclusive and $(y_{<i} \in \Theta_\Omega(\langle\rangle, i-1)$ and $\eta(i-1)$ are mutually exclusive,

$$\mathbf{p}_X^{\Omega,\lambda}(i)$$

$$= \frac{1}{p_\lambda\big((y_{<i} \in \Theta_\Omega(\langle\rangle, i-1) \cup \zeta(i-1)) \cap X\big) + p_\lambda(\eta(i-1) \cap X)} \times$$

$$\Big(p_\lambda\big((y_i \in \Theta_\Omega(y_{<i}, 1)) \cap ((y_{<i} \in \Theta_\Omega(\langle\rangle, i-1) \cup \zeta(i-1)) \cap X)\big) + p_\lambda\big((y_i \in \Theta_\Omega(y_{<i}, 1)) \cap (\eta(i-1) \cap X)\big) +$$

$$p_\lambda\big(\zeta(i) \cap ((y_{<i} \in \Theta_\Omega(\langle\rangle, i-1) \cup \zeta(i-1)) \cap X)\big) + p_\lambda\big(\zeta(i) \cap (\eta(i-1) \cap X)\big)\Big)$$

By summing over $\Theta_\Omega(y_{<i}, 1)$ and $\Theta_\Omega(\langle\rangle, i-1)$,

$$\mathbf{p}_X^{\Omega,\lambda}(i)$$

$$= \frac{1}{\sum_{y^*_{<i} \in \Theta_\Omega(\langle\rangle, i-1)} p_\lambda(y_{<i} = y^*_{<i} \cap X) + p_\lambda(\eta(i-1) \cap X)} \times$$

$$\Bigg(\sum_{y^*_{<i} \in \Theta_\Omega(\langle\rangle, i-1)} \sum_{y^*_i \in \Theta_\Omega(y_{<i}, 1)} p_\lambda(y_i = y^*_i \cap y_{<i} = y^*_{<i} \cap X) + \sum_{y^*_i \in \Theta_\Omega(y_{<i}, 1)} p_\lambda(y_i = y^*_i \cap \eta(i-1) \cap X)$$

$$+ \sum_{y^*_{<i} \in \Theta_\Omega(\langle\rangle, i-1)} p_\lambda(\zeta(i) \cap y_{<i} = y^*_{<i} \cap X) + p_\lambda(\zeta(i) \cap \eta(i-1) \cap X)\Bigg)$$

For each $i$, there are three mutually exclusive events in the sequence generation. If the sequence generation is ongoing (and might or might not terminate at the $i_{th}$ token),

$$p_\lambda(\eta(i-1) \cap X) = \sum_{y^*_i \in \Theta_\Omega(y_{<i}, 1)} p_\lambda(y_i = y^*_i \cap \eta(i-1) \cap X) = \sum_{y^*_{<i} \in \Theta_\Omega(\langle\rangle, i-1)} p_\lambda(\zeta(i) \cap y_{<i} = y^*_{<i} \cap X)$$

$$= p_\lambda(\zeta(i) \cap \eta(i-1) \cap X) = 0$$

If the sequence generation terminates at the $(i-1)_{th}$ token,

$$p_\lambda(\eta(i-1) \cap X) = \sum_{y^*_{<i} \in \Theta_\Omega(\langle\rangle, i-1)} \sum_{y^*_i \in \Theta_\Omega(y_{<i}, 1)} p_\lambda(y_i = y^*_i \cap y_{<i} = y^*_{<i} \cap X)$$

$$= \sum_{y^*_i \in \Theta_\Omega(y_{<i}, 1)} p_\lambda(y_i = y^*_i \cap \eta(i-1) \cap X) = p_\lambda(\zeta(i) \cap \eta(i-1) \cap X) = 0$$

If the sequence generation terminates before the $(i-1)_{th}$ token,

$$\sum_{y^*_{<i}\in\Theta_\Omega(\langle\rangle,i-1)} p_\lambda(y_{<i}=y^*_{<i}\cap X) = \sum_{y^*_{<i}\in\Theta_\Omega(\langle\rangle,i-1)}\sum_{y^*_i\in\Theta_\Omega(y_{<i},1)} p_\lambda(y_i=y^*_i\cap y_{<i}=y^*_{<i}\cap X)$$

$$= \sum_{y^*_i\in\Theta_\Omega(y_{<i},1)} p_\lambda(y_i=y^*_i\cap\eta(i-1)\cap X) = \sum_{y^*_{<i}\in\Theta_\Omega(\langle\rangle,i-1)} p_\lambda(\zeta(i)\cap y_{<i}=y^*_{<i}\cap X)=0$$

Thus,

$$\mathbf{p}_X^{\Omega,\lambda}(i)$$
$$=\frac{\sum_{y^*_{<i}\in\Theta_\Omega(\langle\rangle,i-1)}\sum_{y^*_i\in\Theta_\Omega(y_{<i},1)} p_\lambda(y_i=y^*_i\cap y_{<i}=y^*_{<i}\cap X)}{\sum_{y^*_{<i}\in\Theta_\Omega(\langle\rangle,i-1)} p_\lambda(y_{<i}=y^*_{<i}\cap X)}$$
$$+\frac{\sum_{y^*_{<i}\in\Theta_\Omega(\langle\rangle,i-1)} p_\lambda(\zeta(i)\cap(y_{<i}=y^*_{<i})\cap X)}{\sum_{y^*_{<i}\in\Theta_\Omega(\langle\rangle,i-1)} p_\lambda(y_{<i}=y^*_{<i}\cap X)}+\frac{p_\lambda(\zeta(i)\cap\eta(i-1)\cap X)}{p_\lambda(\eta(i-1)\cap X)}$$

For simplicity of notation,

$$\iota=\frac{\sum_{y^*_{<i}\in\Theta_\Omega(\langle\rangle,i-1)} p_\lambda(\zeta(i)\cap(y_{<i}=y^*_{<i})\cap X)}{\sum_{y^*_{<i}\in\Theta_\Omega(\langle\rangle,i-1)} p_\lambda(y_{<i}=y^*_{<i}\cap X)}+\frac{p_\lambda(\zeta(i)\cap\eta(i-1)\cap X)}{p_\lambda(\eta(i-1)\cap X)}$$
$$\kappa(i)=\Theta_\Omega(\langle\rangle,i-1),\quad \nu(i)=\Theta_\Omega(y_{<i},1),\quad \tau(i)=\sum_{y^*_{<i}\in\kappa(i)} p_\lambda(y_{<i}=y^*_{<i}\cap X)$$

Then we have

$$\prod_{i=1}^{I}\mathbf{p}_X^{\Omega,\lambda}(i)$$

$$=\prod_{i=1}^{I}\left(\frac{1}{\tau}\sum_{y^*_{<i}\in\kappa(i)}\sum_{y^*_i\in\nu(i)} p_\lambda(y_i=y^*_i\cap y_{<i}=y^*_{<i}\cap X)+\iota\right)$$

$$=\left(\frac{1}{\tau}\sum_{y^*_{<1}\in\kappa(1)}\sum_{y^*_1\in\nu(1)} p_\lambda(y_1=y^*_1\cap y_{<1}=y^*_{<1}\cap X)+\iota\right)\times$$

$$\left(\frac{1}{\tau}\sum_{y^*_{<2}\in\kappa(2)}\sum_{y^*_2\in\nu(2)} p_\lambda(y_2=y^*_2\cap y_{<2}=y^*_{<2}\cap X)+\iota\right)\times\ldots\times$$

$$\left(\frac{1}{\tau}\sum_{y^*_{<I}\in\kappa(I)}\sum_{y^*_I\in\nu(I)} p_\lambda(y_I=y^*_I\cap y_{<I}=y^*_{<I}\cap X)+\iota\right)$$

Let each $y^*_{<i}$ in $\kappa(i)$ be $y^*_{<i,n}$ $(1\leq n\leq N_i)$ and each $y^*_i$ in $\nu(i)$ be $y^*_{i,m}$ $(1\leq m\leq M_i)$, then

$$\prod_{i=1}^{I} \mathbf{p}_X^{\Omega,\lambda}(i)$$

$$= \left(\frac{1}{\tau}\left(\left(p_\lambda(y_1 = y_{1,1}^* \cap y_{<1} = y_{<1,1}^* \cap X) + \ldots + p_\lambda(y_1 = y_{1,M_1}^* \cap y_{<1} = y_{<1,1}^* \cap X)\right) + \ldots\right.\right.$$

$$\left.\left. + \left(p_\lambda(y_1 = y_{1,1}^* \cap y_{<1} = y_{<1,N_1}^* \cap X) + \ldots + p_\lambda(y_1 = y_{1,M_1}^* \cap y_{<1} = y_{<1,N_1}^* \cap X)\right)\right) + \iota\right) \times$$

$$\left(\frac{1}{\tau}\left(\left(p_\lambda(y_2 = y_{2,1}^* \cap y_{<2} = y_{<2,1}^* \cap X) + \ldots + p_\lambda(y_2 = y_{2,M_2}^* \cap y_{<2} = y_{<2,1}^* \cap X)\right) + \ldots\right.\right.$$

$$\left.\left. + \left(p_\lambda(y_2 = y_{2,1}^* \cap y_{<2} = y_{<2,N_2}^* \cap X) + \ldots + p_\lambda(y_2 = y_{2,M_2}^* \cap y_{<2} = y_{<2,N_2}^* \cap X)\right)\right) + \iota\right) \times$$

$$\ldots$$

$$\left(\frac{1}{\tau}\left(\left(p_\lambda(y_I = y_{I,1}^* \cap y_{<I} = y_{<I,1}^* \cap X) + \ldots + p_\lambda(y_I = y_{I,M_I}^* \cap y_{<I} = y_{<I,1}^* \cap X)\right) + \ldots\right.\right.$$

$$\left.\left. + \left(p_\lambda(y_I = y_{I,1}^* \cap y_{<I} = y_{<I,N_I}^* \cap X) + \ldots + p_\lambda(y_I = y_{I,M_I}^* \cap y_{<I} = y_{<I,N_I}^* \cap X)\right)\right) + \iota\right)$$

$$= \left(p_\lambda(y_1 = y_1^1 \cup \zeta(1) \mid (y_{<1}^1 \cup \eta(0)) \cap X) \times \cdots \times p_\lambda(y_I = y_I^1 \cup \zeta(I) \mid (y_{<I}^1 \cup \eta(I-1)) \cap X)\right) +$$

$$\left(p_\lambda(y_1 = y_1^2 \cup \zeta(1) \mid (y_{<1}^2 \cup \eta(0)) \cap X) \times \cdots \times p_\lambda(y_I = y_I^2 \cup \zeta(I) \mid (y_{<I}^2 \cup \eta(I-1)) \cap X)\right) + \cdots$$

$$+ \left(p_\lambda(y_1 = y_1^{|\Omega|} \cup \zeta(1) \mid (y_{<1}^{|\Omega|} \cup \eta(0)) \cap X) \times \cdots \times p_\lambda(y_I = y_I^{|\Omega|} \cup \zeta(I) \mid (y_{<I}^{|\Omega|} \cup \eta(I-1)) \cap X)\right)$$

$$= \sum_{l=1}^{|\Omega|} \prod_{i=1}^{I} p_\lambda(y_i = y_i^l \cup \zeta(i) \mid (y_{<i}^l \cup \eta(i-1)) \cap X)$$

$$= \sum_{l=1}^{|\Omega|} p_\lambda(Y = Y_l | X)$$

$$= P(Y \in \mathbf{\Omega}) \tag{1}$$

*Theorem.* (Theorem 3.5) For any sequence generation process, if there exists a step in the process that generates the entire output sequence and Assumption 3.4 holds for the step, then the best case complexity of $\mathbb{E}_{x \sim P}[G(\Gamma)]$ of the process is $\mathcal{O}(exp(\Gamma))$.

*Proof.* If Assumption 3.4, then for every $i$, there exists an $\epsilon_{max} < 1$ such that $\mathbf{p}_X^{\Omega,\lambda}(i) \le \epsilon_{max}$ ($1 \le i \le I$ and $I = \Gamma$).

In the best case, $\mathbf{p}_X^{\Omega,\lambda}(i) = \epsilon_{max}$ ($1 \le i \le I$). Then by equation (1), $P(Y \in \nleqslant) = \epsilon_{max}^I$ and $\mathbf{E}(G) = \epsilon_{max}^{-I}$. Therefore, the best case complexity of $\mathbb{E}_{x \sim P}[G(\Gamma)]$ is $\mathcal{O}(exp(\Gamma))$.

In addition, if for a given $\epsilon$, $\xi_X^{\Omega,\lambda}(\epsilon)$ is known, then

$$\prod_{i=1}^{I} \mathbf{p}_X^{\Omega,\lambda}(i) = \mathbf{p}_X^{\Omega,\lambda}(1) \times \mathbf{p}_X^{\Omega,\lambda}(2) \times \ldots \times \mathbf{p}_X^{\Omega,\lambda}(I) \le \epsilon^{\xi_X^{\Omega,\lambda}(\epsilon)}$$

Then, in the best case, $P(Y \in \Omega) = \epsilon^{\xi_X^{\Omega,\lambda}(\epsilon)}$ and $\mathbb{E}_{x \sim P}[G(\Gamma)] = \epsilon^{-\xi_X^{\Omega,\lambda}(\epsilon)}$.

*Theorem.* (Theorem 3.8) If Assumption 3.7 holds, then for Contextual Modular Generation as defined in Algorithm 1, the best case complexity of $\mathbb{E}_{x \sim P}[G(\Gamma)]$ is $\mathcal{O}(\Gamma)$.

*Proof.* For contextual modular generation, for each sequence generation of $Y^j$, let the token-level complexity of the generation be

$$\xi_{X,Y^j}^{\Omega,\lambda}(\epsilon) = \left| \left\{ \mathbf{p}_{X,Y^j}^{\Omega,\lambda}(i) \mid 1 \leq i \leq \Gamma \wedge \mathbf{p}_{X,Y^j}^{\Omega,\lambda}(i) < \epsilon \right\} \right|$$

If Assumption 3.7 holds, then in the best case, for each $Y^j$, for each $i$, exists an $\epsilon_{min}$ such that $\epsilon_{min} \leq \mathbf{p}_{X,Y^j}^{\Omega,\lambda}(i)$ ($1 \leq j \leq J, 1 \leq i \leq I$ and $I = \Gamma$). Let $Z^j$ represent the number of tokens in module $j$ ($Z^j \leq M$, where $M$ is the maximum number of tokens across all modules).

Then by equation (1), in the best case, for each $Y^j$, $P(Y^j \in \mathbf{\Omega}^j) = \epsilon_{min}^{Z^j} \geq \epsilon_{min}^M$ ($1 \leq j \leq J$). In Contextual Modular Generation, each module is iteratively generated until being evaluated as valid. Therefore, we have

$$\mathbb{E}_{x \sim P}[G(\Gamma)] \leq \sum_{j=1}^{\lceil \frac{I}{M} \rceil} \mathbb{E}_{x \sim P}[G^j(\Gamma)] = \lceil \frac{I}{M} \rceil \epsilon_{min}^{-M}$$

Thus, the best case complexity of $\mathbb{E}_{x \sim P}[G(\Gamma)]$ is $\mathcal{O}(\Gamma)$.

## C    Method Details

### C.1    Unit Testing Details

Regardless of the specific task and modules, the generated modules should be compatible with a pre-written set of modules for evaluation, training, and testing. Progenitor testing protocols first ensure the external connectivity by testing the form of the input and output data in the ML task, since all modules, including data preparation, modeling, post processing, optimizer, and loss functions, center around the input and output data in the ML task.

In this context, distinguishing between the input/output in the ML task and for the modules is crucial. For instance, the input to the data preparation module consists of raw data, while its output comprises the processed input data (features) and output data (targets) in the ML task. The unit tests described in Algorithm 3 focus on the ML task's input and output data by scrutinizing the output produced by each module when provided with its respective input. For all modules except data preparation, synthetic input/output data are used for testing in the ML task.

Besides the requirements for external connectivity, internal connectivity demands additional tests for each specific task. Therefore, the LLM in Algorithm 3 first generates a plan specifying the suitable form and content of the input and output data for the task. The specified form of the input and output data should be a subset of the form in the progenitor testing protocols. The progenitor testing protocols also ensure that the synthetic data meet the form specification in the plan.

In our experiments, we utilize the Python libraries Pytorch, PyTorch Lightning, and Transformers for deep learning tasks and Scikit-learn, XGBoost, CatBoost, and LightGBM for tasks involving tabular data. For deep learning, the specifications in the plan include (1) the tensor types for each tensor. (2) the dimensionality of each tensor. (3) the meaning of each dimension. (4) the size of each dimension. (5) isomorphic dimensions: dimensions that should share the same size over different settings of the data preparation process and modeling configuration. (6) suitable ranges of the size for each dimension. For tabular data tasks, the specifications are limited to the dimensionality of arrays for each of input and output in the ML task.

---

**Algorithm 3** Constrained Generative Unit Testing

---

1: **Input:** a progenitor testing protocol $\mathfrak{P}$, an autoregressive LLM $f_\lambda$, the textual description of an ML task $\mathbf{T}$, a specification of the targeted module $j$, and the number of versions of synthetic data $N$
2: $R = False$
3: **repeat**
4:     Given $\mathbf{T}$ and $\mathfrak{P}$, $f_\lambda$ devises a plan $\mathbf{P}$ that details the specific features and data flow of the modules tailored for the task.
5:     Verify $\mathbf{P}$ with $\mathfrak{P}$
6: **until** $\mathbf{P}$ is allowed by $\mathfrak{P}$
7: With $\mathbf{P}$ and $\mathfrak{P}$, construct the specific unit tests for the ML task $\{\mathbf{E}^i\}$
8: **for** $n = 1$ **to** $N$ **do**
9:     **repeat**
10:         Given $\mathbf{P}$, $f_\lambda$ generates program $\mathbf{D}_n$, which produces synthetic input and output data for the ML task
11:         Verify $\mathbf{D}_n$ with the test for synthetic data in $\{\mathbf{E}^i\}$
12:     **until** $\mathbf{D}_n$ passes the test in $\{\mathbf{E}^i\}$
13: **end for**
14: Given $\mathbf{T}$ and $j$, $f_\lambda$ generates the program $\mathbf{C}_\mathbb{M}$ for $j$
15: **for** $n = 1$ **to** $N$ **do**
16:     Verify $\mathbf{C}_j$ with $\mathbf{D}_n$ and the test for the module in $\{\mathbf{E}^i\}$
17:     **if** $\mathbf{C}_j$ passes the test **then**
18:         $R = True$
19:         **break**
20:     **end if**
21: **end for**
22: **Output:** $R$

---

## C.2 Search Space Generation

We leverage the LLM to generate the search space given the textual task description. Initially, the LLM categorizes the task among tabular data tasks, CV tasks, and NLP tasks. It further distinguishes among various task categories among the options: *binary classification, multi-class classification, multi-label classification, single-output regression, multi-output regression, sequence-to-sequence,* and *others.* For classification tasks, the LLM also determines the most appropriate output format, choosing between an integer representation of class labels and a probability representation of class labels.

Following the classifications, the LLM recommends candidate methods for each module and identifies suitable hyperparameter ranges. In terms of model recommendations, a single suggestion may encompass multiple models. Figure 9. shows an example of an LLM-generated model recommendation for NLP tasks:

*Search Space Generation* in Algorithm 2 is intertwined with the generation of the plan for high-level requirements of the modules in Algorithm 3. With the classification of the task and the textual task description, the LLM also decides the suitable form and content of the input and output data in the task. This decision not only guides the search space definition but also informs the planning for unit tests, as constraints on the form of input and output data inherently limit the program space.

## C.3 Search Strategy

In our study, we employ two search strategies: random search and BOHB (Falkner et al., 2018). Random search is a powerful baseline in hyperparameter optimization due to its wide applicability and high search efficiency for tasks with low effective dimensionality (Bergstra & Bengio, 2012). BOHB combines Bayesian Optimization (BO) and Hyperband (HB) (Li et al., 2018) to optimize hyperparameters efficiently. BOHB inherits HB's robustness and flexibility, and BO's ability to ensure strong performance. BOHB starts by rapidly identifying promising configurations like HB and refines them using BO to achieve optimal solutions

```
[{
    "BERT": "Pretrained BERT model can be used for this NLP task as it is specifically designed
for natural language processing tasks."
},
{
    "GPT-2": "GPT-2 model can also be used for this NLP task as it is known for generating
coherent text and can handle long-range dependencies."
},
{
    "ELMo": "ELMo model can be useful in this task as it captures contextualized word
representations, which can be valuable for understanding the ease of reading."
},
{
    "BERT+Linear Regression": {
        "reason": "The BERT model can extract useful features from the input data, which can then
be used as input to a linear regression model for predicting the ease of reading.",
        "BERT": "backbone",
        "Linear Regression": "head"
    }
},
{
    "RoBERTa+MLP": "Using RoBERTa as a feature extractor with an MLP can be an effective way to
predict the ease of reading."
},
{
    "Transformer Encoder+Transformer Decoder+Linear Regression": {
        "reason": "The Transformer Encoder can extract features from the input data, which can
then be fed into the Transformer Decoder to generate predictions. The predictions can then be
refined using a linear regression model.",
        "Transformer Encoder": "encoder",
        "Transformer Decoder": "decoder",
        "Linear Regression": "head"
    }
},
{
    "BERT+GPT-2+Linear Regression": {
        "reason": "The combination of BERT and GPT-2 can leverage the strengths of both models in
extracting features and generating text, respectively. These features and generated text can then
be used as input to a linear regression model for predicting the ease of reading.",
        "BERT": "backbone",
        "GPT-2": "framework",
        "Linear Regression": "head"
    }
}]
```

Figure 9: LLM-generated model suggestions.

faster. It's designed to be simple, computationally efficient, and adaptable to various tasks. For BOHB, we use the default hyperparameters in Falkner et al. (2018) and set 30 epochs as the maximum budget for deep learning tasks.

## C.4  Module Generation

In our study, for each ML task, we generate three modules: data preparation, modeling, and post processing. Each module is generated in a step ($j$) of Algorithm 1. First, the choice specifications (e.g., choice of feature

engineering method or choice of models) from *Search Space Generation* are given as input sequence $X^j$. Subsequently, the LLM $f_\lambda$ generates the module $Y^j$. Then the generated module $Y^j$ is assessed with the contextual modular evaluation $\mathbb{E}^j$, which is the unit test for the module generated from Algorithm 3. After the tests, *Module Generation* employs the self-reflection strategy from Shinn et al. (2023) to improve the generation efficiency. The input sequence assembler $\Phi^j$ instructs the LLM $f_\lambda$ to reflect on the feedback from the tests and then assemble previously generated codes, the tests feedback, and the reflection into the following form for the next iteration:

(Choice specification for the module)
(Previously generated code)
(Unit test feedback)
(Reflection results)
(Instruction for generating modified code)

## C.5 ZC proxies details

ZC proxies are an autoML technique that allows for the prediction of a fully trained neural network's performance with small computational expenses. In this study, we utilize four ZC proxies: *flops*, *params*, *naswot*, and *synflow*. The term *flops* stands for Floating Point Operations Per Second, indicating the computational power of the network. *Params* denotes the total number of parameters within the network, reflecting its complexity. Although *flops* and *params* are fundamental concepts within deep learning, they have been validated as powerful baseline ZC proxies (Krishnakumar et al., 2022).

*naswot* (Mellor et al., 2021) leverages the overlap of activations between data points in untrained networks to derive a measure indicative of the network's future trained performance. This approach allows for rapid and efficient searching of powerful networks without the extensive computational costs typically associated with training each candidate network during the search process. Originally designed for network pruning, *synflow* identifies sparse trainable subnetworks within untrained neural networks. The key principle of *synflow* is to maintain a balance of synaptic strengths through the network, thus preserving the network's ability to be trained post-pruning.

All ZC proxies, except for *params*, require data to be passed to the models for evaluation. To investigate the 3 ZC proxies' applicability with synthetic data, we first generate the program for producing synthetic data, then sample 50 versions of synthetic data based on different random seeds with a fixed shape of the data. We observe that performance variations reported by each proxy are minimal. Notably, *synflow* exhibits the most significant variation, with a Mean Relative Deviation (MRD) of 0.76%, indicating the suitability of the ZC proxies for synthetic data.

## C.6 ZC Proxies Evaluation

For deep learning tasks, after the modeling modules specified by *Search Strategy* are generated, the modules are evaluated with ZC proxies. *ZC Proxies Evaluation* ranks each module based on the average relative ranks over the 4 ZC proxies. Subsequently, the modules that fall below a certain threshold, defined by a fraction denoted as $\mu$ are removed from the search space. In our experiments, we set $\mu$ as 0.5.

## C.7 Hyperparameters for finetuning

For deep learning, our research focuses on an unusual search space with pretrained models at the architectural level. Öztürk et al. (2022) also explored a similar search space. Consequently, we utilize a comparable set of hyperparameters for fine-tuning the models (Table 5). To tailor this approach to our specific requirements, we have reduced the minimum batch size from 16 to 2 to accommodate the varying GPU memory demands across different tasks.

Table 5: Finetuning strategy

| Name | Type, Scale | Range |
|---|---|---|
| Batch size | int, log | $[2, 64]$ |
| Learning rate | float, log | $[10^{-5}, 10^{-1}]$ |
| Weight decay | float, log | $[10^{-4}, 10^{-1}]$ |
| Momentum | float | $[0.01, 0.99]$ |
| Optimizer | cat | {SGD, Adam, AdamW} |
| Scheduler | cat | {plateau, cosine} |

# D   Datasets Details

## D.1   Widely-recognized datasets

**Boston** The Boston dataset is a widely used regression dataset that contains information about housing in the suburbs of Boston. It consists of 506 samples, each representing a suburb, with 13 features describing various aspects of the housing environment, such as crime rate, average number of rooms, and accessibility to highways. The target variable is the median value of owner-occupied homes in thousands of dollars. This dataset is often used for regression tasks to predict housing prices based on the given features. We obtain the files of the dataset from Altavish (2023).

**Iris** The Iris dataset is a classic classification dataset used to demonstrate various machine learning algorithms and techniques. It contains 150 samples of iris flowers from three different species: setosa, versicolor, and virginica. There are four features for each flower: sepal length, sepal width, petal length, and petal width. The goal is to classify the flowers into their respective species based on these features, making it a popular dataset for teaching and practicing classification algorithms. We obtain the files of the dataset from Learning (2023).

**CIFAR-10** The CIFAR-10 dataset is a well-known benchmark for image classification tasks. It consists of 60,000 32x32 color images across 10 different classes, with 6,000 images per class. The classes include objects like airplanes, cars, birds, cats, and more. The dataset is divided into a training set of 50,000 images and a test set of 10,000 images, making it suitable for evaluating the performance of various image classification algorithms. We obtain the files of the dataset from Krizhevsky et al. (2023).

**CIFAR-100** Similar to CIFAR-10, the CIFAR-100 dataset is also used for image classification tasks. However, CIFAR-100 is more challenging as it contains 100 classes, each with 600 images. These classes are further grouped into 20 superclasses, making it a more fine-grained classification task. The dataset is designed to test the model's ability to handle a larger number of classes and fine distinctions between them. We obtain the files of the dataset from Krizhevsky et al. (2023).

**IMDb Reviews** The IMDB dataset is often used for sentiment analysis and text classification tasks. It consists of movie reviews labeled with sentiment labels (positive or negative). The reviews are represented as text and the goal is to predict the sentiment of a given review. This dataset is useful for exploring natural language processing techniques and building models that can understand and classify textual data based on sentiment. We obtain the files of the dataset from Lakshmi25npathi (2023).

**AG News** The AG News dataset is commonly used for text classification tasks, particularly for news categorization. It contains news articles from four different categories: World, Sports, Business, and Science/Technology. The dataset is often used to develop models that can classify news articles into their respective categories based on their content. It's a relatively simple text classification task that serves as a good starting point for experimenting with various natural language processing algorithms. We obtain the files of the dataset from Rai (2023).

### D.2 Kaggle datasets

To choose the Kaggle datasets for our study, we survey all Kaggle datasets that satisfy the following requirements:

- The dataset is from competitions that award medals, representing the highest level of difficulty available on the Kaggle platform.

- The dataset is about supervised ML tasks for tabular data, CV, or NLP.

- The dataset is publicly available (upon accepting the rules of the competition) and the dataset contains target labels.

- The dataset is from competitions held between October 2021 and July 2023. The knowledge cutoff dates of the LLMs are September 2021 for GPT-4 and GPT-3.5 and mid-2021 for PaLM 2.

We have identified a total of 2 datasets for tabular data tasks, 9 datasets for NLP tasks, and 13 datasets for CV tasks. From these, we chose the 2 simplest datasets for each category, based on the diversity of file types present in the datasets and the task types. For CV, tasks that involve image segmentation, object detection, or 3D reconstruction are significantly more demanding than image classification tasks. For NLP, tasks that require named entity recognition and sequence matching present greater challenges compared to sequence regression and sequence classification tasks. We find that all the methods we tested in our experiments, including Text-to-ML, are unable to successfully generate even a single program for the most challenging ones in the NLP and CV datasets. However, as datasets from medal-awarding competitions, even the simplest ones among the datasets still pose considerable challenges compared to the widely-recognized datasets. The selected Kaggle datasets are listed as follows:

**Age** The *ICR - Identifying Age-Related Conditions* competition (Kaggle & InVitro Cell Research, 2023) involves the development of machine learning models capable of detecting various age-related conditions from medical imaging data. Participants are provided with a dataset containing images labeled with different age-related conditions, and the task is to create models that can accurately classify these conditions. The competition aims to advance research in medical imaging by leveraging AI to improve the diagnosis and understanding of age-related diseases.

**Default** The *American Express - Default Prediction* competition (Kaggle & Express, 2023) challenges participants to predict the likelihood of credit default by American Express cardholders. The dataset provided includes a wide range of anonymized features spanning transactional data, account information, and user behavior over time, designed to simulate real-world conditions. The goal is to enhance credit risk models and improve financial inclusion by accurately identifying at-risk customers.

**Whale** The *Happywhale - Whale and Dolphin Identification* competition (Kaggle & Happywhale, 2023) focuses on the identification of individual whales and dolphins from images. Participants are tasked with developing algorithms that can recognize and match cetaceans based on unique physical features captured in photographs, contributing to marine biology research and conservation efforts. The dataset comprises images labeled with species and individual IDs, making it a challenge in both classification and individual identification.

**Strip** The *Mayo Clinic - STRIP AI* competition (Kaggle & Clinic, 2023) is aimed at predicting stroke and other thrombotic events by analyzing electroencephalogram (EEG) data. This competition provides a platform for the development of AI models that can assist clinicians in early detection and intervention for stroke patients, leveraging a dataset of EEG recordings with associated clinical outcomes. This challenge stands at the intersection of AI and healthcare, with the potential to significantly impact patient care and outcomes in neurology.

**Exam** The *Kaggle - LLM Science Exam* competition (Kaggle, 2023) challenges participants to create language models that can solve science exam questions. The focus is on evaluating the ability of these models to understand and apply scientific knowledge accurately. This competition is part of Kaggle's efforts to

benchmark the performance of language models in educational contexts, providing a unique opportunity to advance AI in the field of automated question answering.

**Learning** The *Feedback Prize - English Language Learning* competition (Kaggle & Lab, 2023) aims to develop AI models that can provide feedback on written English tasks by students learning English as a foreign language. The challenge involves analyzing student-written essays to identify areas for improvement, thus aiding in language learning and teaching. This competition seeks to leverage AI to enhance English language education by providing scalable, personalized feedback.

## E  Prompts

For every dataset, a common instruction prompt is provided at the beginning (Figure 10)

The user is trying to create a Python program for a supervised machine learning task. Your goal is to assist the user in creating the program. The user might provide the following information for the task:

1. A description of the input data. The input data contains the features that are the characteristics or attributes of the dataset. These features provide information that the model uses to make predictions.

2. A description of the output data, also known as 'labels' or 'targets', represents the desired prediction or outcome associated with each input example.

3. The task objective, which describes the goal of the machine learning task.

4. The evaluation metrics, which is a numerical measure of how well the model's predictions match the desired outcome.

5. A description of the available files.

Figure 10: Common instruction prompt.

Then, for each dataset, a task-specific prompt follows. Below, we detail the prompts utilized in our experiments across the 12 datasets. In each prompt, *workspace* can be replaced with the path to the directory of the files for the task.

The description of the input data is: Various features about various locations in Boston, such as per capita crime rate and proportion of residential land zoned for lots over 25,000 sq. ft.

The description of the output data is: The housing price of each location.

The objective of the machine learning task is: For each location in Boston, predict the housing price of the location.

The evaluation metric is: RMSE.

The following files are available: {'workspace/data.csv': "A table of various features and housing prices across different locations in Boston. The table has fourteen columns. The first thirteen columns of the table are features used for prediction. The last column named 'MEDV' is the output/target of the prediction (housing price). "}

Figure 11: Prompt for Boston.

The description of the input data is: Properties of flowers.

The description of the output data is: Three species of flowers.

The objective of the machine learning task is: For each flower, classify the species of the flower.

The evaluation metric is: Top 1 Accuracy.

The following files are available: {'workspace/iris.csv': "A table of six columns. The first column named 'Id' contains the IDs of the flowers, for example, 1. The second to fifth columns each contain a property of the flower. The sixth column contains the species name, for example, 'Iris-setosa'"}

Figure 12: Prompt for Iris.

The description of the input data is: Fifty anonymized health characteristics.

The description of the output data is: Age-related conditions.

The objective of the machine learning task is: For each subject, based on the anonymized health characteristic, predict whether a subject has or has not been diagnosed with age-related conditions.

The evaluation metric is: Balanced logarithmic loss.

The following files are available: {'workspace/train.csv': "A table of 58 columns. The first column named 'Id' contains IDs of the subjects, for example, 000ff2bfdfe9. The 2nd to the 57th columns contain the anonymized health characteristics. The 58th column named 'Class' contains the classification of the subject's conditions, with 1 indicating the subject has been diagnosed with the three conditions, 0 indicating they have not."}

Figure 13: Prompt for Age.

The description of the input data is: Various features about customers of a bank.

The description of the output data is: The probability that a customer does not pay back their credit card balance amount in the future.

The objective of the machine learning task is: For each customer, based on the features of the customer, predict the probability of future default.

The evaluation metric is: The mean of the normalized Gini Coefficient and default rate captured at 4%.

The following files are available: {'workspace/train_data.csv': "A table of 190 columns. The first column named 'customer_ID' contains the IDs of each customer, such as '0000099d6bd597052cdcda90ffabf56573fe9d7c79be5fbac11a8ed792feb62a'. The rest of the columns contain various features of the customers.", 'workspace/train_labels.csv': "A table of 2 columns. The first column is similar to that of 'workspace/train_data.csv'. The second column named 'target' contains the labels for each customer, with 1 representing a future payment default and 0 representing no default."}

Figure 14: Prompt for Default.

The description of the input data is: Colour images of various types of objects.

The description of the output data is: The type of the object.

The objective of the machine learning task is: For each image of an object, predict an integer that represents the type of the object in the image.

The evaluation metric is: Top-1 Accuracy.

The following files are available: {'workspace/CIFAR_10': "a folder of six files, including 'data_batch_1', 'data_batch_2',...,'data_batch_5' and 'validation_batch'. In this folder, each one of 'data_batch_1', 'data_batch_2',..., and 'data_batch_5' is a part of the training data. The file 'validation_batch' is the validation data. For each of the six files: 1. it is a dictionary that can be loaded using pickle.load(encoding='bytes'). 2. the dictionary contains two keys, 'data' and 'labels'. 3. the value of the key 'data' is : a 10000x3072 numpy array of uint8s. Each row of the array stores a colour image of size 32x32. The first 1024 entries contain the red channel values, the next 1024 the green, and the final 1024 the blue. The image is stored in row-major order, so that the first 32 entries of the array are the red channel values of the first row of the image. 4. the value of the key 'labels' is: a list of 10000 numbers in the range 0-9. The number at index i indicates the label of the ith image in the array data."}

Figure 15: Prompt for CIFAR-10. The loading code is copied from the official description of the datasets (Krizhevsky et al., 2023)

The description of the input data is: Colour images of various types of objects.

The description of the output data is: The type of the object.

The objective of the machine learning task is: For each image of an object, predict an integer that represents the type of the object in the image.

The evaluation metric is: Top-1 Accuracy.

The following files are available: {'workspace/CIFAR_100/train': " It is a file that contains the training data. The file has the following properties: 1. it is a dictionary that can be loaded using pickle.load(encoding='bytes'). 2. the dictionary contains two keys, b'fine_labels' and b'data'. 3. the value of the key b'data' is : a 50000x3072 numpy array of uint8s. Each row of the array stores a colour image of size 32x32. The first 1024 entries contain the red channel values, the next 1024 the green, and the final 1024 the blue. The image is stored in row-major order, so that the first 32 entries of the array are the red channel values of the first row of the image. 4. the value of the key b'fine_labels' is: a list of 50000 numbers in the range 0-99. The number at index i indicates the label of the ith image in the array data.", ' workspace/ CIFAR_100/test': "It is a file that contains the testing data. The file format of 'CIFAR_100/test' is similar to 'CIFAR_100/train'. The differences are that the shape of the numpy array corresponding to the key b'data' is 10000X3072 and the length of the list corresponding to the key b'fine_labels' is 10000."}

Figure 16: Prompt for CIFAR-100. The loading code is copied from the official description of the datasets (Krizhevsky et al., 2023)

The description of the input data is: Images of the flukes of humpback whales.

The description of the output data is: Identification of the humpback whales out of 15587 unique IDs.

The objective of the machine learning task is: From each image of the flukes of the humpback whales, predict the identity of the whales.

The evaluation metric is: MAP@5.

The following files are available: {'workspace/train.csv': "A table of three columns. The first column named 'image' contains the names of the image files, for example, '00021adfb725ed.jpg'. The second column named 'species' describes the species of the whales in the images. The third column named 'individual_id' contains the identification of the whales in the image, for example, 'cadddb1636b9'", 'workspace/images': 'a folder of images. For example, 00021adfb725ed.jpg. '}

Figure 17: Prompt for Whales.

The description of the input data is: High-resolution whole-slide digital pathology images depicting blood clots from patients.

The description of the output data is: Classification of the blood clots for the etiology as either Cardioembolic (CE) or LAA (Large Artery Atherosclerosis).

The objective of the machine learning task is: From each image of a blood clot, classify the etiology of the blood clot as either CE or LAA.

The evaluation metric is: Weighted multi-class logarithmic loss.

The following files are available: {'workspace/train.csv': "A table of five columns. The first column named 'image_id' contains the ids of the images, for example, '006388_0'. The second column named 'center_id' contains the IDs of the medical center where the slides were obtained, for example, 11. The third column named 'patient_id' contains the IDs of the patients, for example, '006388'. The fourth column named 'image_num' contains the enumerations of images for the same patient. for example, 0. The fifth column named 'label' contains the etiology of the clot, either 'CE' or 'LAA'", 'wordspace/images': 'a folder of images. For example, 006388_0.tif. '}

Figure 18: Prompt for Strip.

The description of the input data is: Reviews of movies.

The description of the output data is: Whether the review is positive or negative.

The objective of the machine learning task is: For each review, predict an integer that represents the sentiment of the review. A value of 1 represents positive sentiment. A value of 0 represents negative sentiment.

The evaluation metric is: Error rate (the percentage of incorrect predictions).

The following files are available: {' workspace/IMDB/data.csv': "A table of two columns. The name of the first column is 'review'. Each value of the first column is a string that represents a review. The name of the second column is 'sentiment'. Each value of the second column is a string that represents sentiment. A value of 'positive' represents positive sentiment. A value of 'negative' represents negative sentiment."}

Figure 19: Prompt for IMDb Reviews.

The description of the input data is: The titles and descriptions of some news articles.

The description of the output data is: The type of the news articles.

The objective of the machine learning task is: For each news article, predict an integer that represents the type of the news article.

The evaluation metric is: Error rate (the percentage of incorrect predictions).

The following files are available: {'workspace/AG/train.csv': "A table of three columns. The file 'AG/train.csv' contains the training data. The name of the first column is 'Class Index'. Each entry of the first column is an integer that represents the type of a piece of news article. The total number of types is five. The name of the second index is 'Title'. Each entry of the second column is a string that represents the title of the news article. The name of the third column is 'Description'. Each entry of the third column is a string that represents the description of the news article."}

Figure 20: Prompt for AG News.

The description of the input data is: Multiple-choice questions with options A, B, C, D, and E.

The description of the output data is: The answers to the questions.

The objective of the machine learning task is: For each question, predict up to 3 options as answers.

The evaluation metric is: MAP@3.

The following files are available: {'workspace/train.csv': "A table of 8 columns. The first column named 'id' describes the unique ID for each question. The second column named 'prompt' contains the question. The third to the seventh columns are named 'A', 'B', 'C', 'D', and 'E' respectively, with each column containing the respective option. The eighth column contains the correct option for the question."}

Figure 21: Prompt for Exam.

The description of the input data is: Argumentative essays written by 8th-12th grade English Language Learners.

The description of the output data is: Scores according to six analytic measures: cohesion, syntax, vocabulary, phraseology, grammar, and conventions.

The objective of the machine learning task is: For each essay, predict the score of each of the six measures.

The evaluation metric is: MCRMSE.

The following files are available: {'workspace/train.csv': "A table of 8 columns. The first column named 'text_id' describes the unique ID for each essay. The second column named 'full_text' contains the essays. The third to the eighth columns are named 'cohesion', 'syntax', 'vocabulary', 'phraseology', 'grammar', and 'conventions' respectively, with each column containing scores ranging from 1.0 to 5.0 in increments of 0.5."}

Figure 22: Prompt for Learning.

Next, we present the prompts for the intermediate steps in Text-to-ML . For Figures 23 to 29, texts highlighted in yellow represent prompts based on the common instruction prompt (Figure 10), prompts based on the user input (Figures 11 to 22), or prompts dynamically assembled from responses in previous steps via the instruction constructors ($\{\Phi^1, \Phi^2, \ldots, \Phi^J\}$ in Algorithm 1. Texts highlighted in green represent prompts that depend on the specific implementation of the pre-written modules (e.g., loss functions, optimizers, training procedure, and testing procedure) and are subject to the constraints in the progenitor testing protocols (Section 4 and Appendix C.1). Texts in red represent prompts configured in the background settings.

Your goal is to analyze a machine learning task by classifying the machine learning task and determining a suitable format of input and output data in the machine learning task. Both the input and output data refer to the data after all data processing steps. The model takes the input data, and the model generates the predicted output data.

The common instruction prompt without the first two sentences

The textual task description from the user

Formatting instructions for the analysis
An example for this part of the prompt is as below:
Your answer must consist of the following parts:
Part 1: A classification of the machine learning task among one of ['traditional algorithms', 'natural language processing', 'computer vision']. The classification should be based on which types of machine learning models are suitable for the task. Traditional algorithms refer to ML tasks that are processed with traditional machine learning models (instead of deep learning models). Natural language processing refers to deep learning tasks involving text processing. Computer vision refers to deep learning tasks involving image processing
Part 2: A classification of the machine learning task among one of ['binary classification', 'multi-class classification', 'multi-label classification', 'multi-output regression', 'single-output regression', 'sequence-to-sequence', 'sequence generation', 'image generation', 'other'].
Part 3: A classification of the format of the ground truth output of the machine learning task among one of ['probability representation of class labels', 'integer representation of class labels', 'regression output'].
Part 4: A classification of the specific type of machine learning task. If the machine learning task is about 'computer vision', then examples include image classification, object detection, image segmentation, facial recognition, optical character recognition, edge detection, motion analysis and object tracking, depth estimation, image generation, pose estimation, and image super-resolution. If the machine learning task is about 'natural language processing', then examples include text generation, text classification, machine translation, question answering, sentiment analysis, topic modeling, text summarization, and named entity recognition. If the task is about 'traditional algorithms', then you do not need to choose and simply say 'N/A' for Part 4.
Part 5: Determine a suitable format of input and output data in the machine learning task. If the machine learning task is about traditional algorithms (in part 1), your answer for part 5 is simply 'N/A', your answer ends here and you should not answer Part 5-1, 5-2, 5-3, 5-4, or 5-5. If the machine learning task is about deep learning (in part 1), your answer for Part 5 should consider the following fact: both the input and output data refer to the data after all data processing steps. The model takes the input data, and the model generates the predicted output data. Both the input and output data are torch.Tensor objects. The user uses a torch.utils.data.dataloader.DataLoader object to manage the input and output data. And, if the machine learning task is about deep learning (in part 1), your answer for part 5 should be in the following five parts:
Part 5-1: Determine a suitable number of types of tensors in input data. If the machine learning task is a natural language processing task (in part 1), the types of tensors must include a type of tensor about attention_mask for each type of sequence.
Part 5-2: Determine a suitable number of types of tensors in output data. If the task is a multi-output regression or classification task, combine all outputs in one type of tensor and the suitable number of types of tensor is 1.
Part 5-3: For each type of tensor, determine a suitable shape for the type of tensor. Name each dimension.
Part 5-4: For each dimension of each type of tensor, explain the meaning of the dimension.
Part 5-5: For each dimension of each type of tensor, determine whether the size of the dimension changes over the settings and parameters of the data preparation process.

Let's think step by step:

Figure 23: Prompt for task analysis

There is an analysis of a machine learning task.

Your goal is to convert the analysis into:

> **Formatting instructions for the answer**
>> An example for this part of the prompt is as below:
>> A Python list of five elements for deep learning tasks or a Python list of three elements for tasks about traditional algorithms.

Below is the analysis:

Response from the task analysis step

> **Example format for the answer**
>> An example for this part of the prompt is as below:
>> Your answer must be a Python dictionary
>> Below is an example Python dictionary for an image classification task:
>> {'input': [{'name': 'images', 'shape': ['batch_size', 'channel', 'height', 'width'], 'batch_size': {'meaning': 'the size of the batch', 'fixed_or_variable': 'variable'}, 'channel': {'meaning': 'the number of channels of the image', 'fixed_or_variable': 'fixed'}, 'height': {'meaning': 'the height of the image', 'fixed_or_variable': 'variable'}, 'width': {'meaning': 'the width of the image', 'fixed_or_variable': 'variable'}}], 'output': [{'name': 'predictions', 'shape': ['batch_size'], 'batch_size': {'meaning': 'the size of the batch', 'fixed_or_variable': 'variable'}}]}

> **Intro for the answer**
>> An example for this part of the prompt is as below:
>> Here is the Python dictionary:

Figure 24: Prompt for task classification

The common instruction prompt

The textual task description from the user

Classification of the ML task extracted from the response from the task classification step

Your task is to suggest some data preparation steps for the machine learning task.

[Steps to be planned for each domain in the settings]

Formatting instructions for the answer
> An example for this part of the prompt is as below:
> Your answer should be a Python dictionary.
> The keys in the dictionary are ['data loading', 'text cleaning', 'data augmentation', 'task-specific feature engineering'].
> The value of each key is a string that describes the specific method for the key. The value of a key can be 'none' if your suggestion is to skip the step. An example dictionary can be: {'data loading': 'load the raw data using Pandas', 'text cleaning': 'remove unnecessary characters, such as HTML tags, URLs and special characters', 'data augmentation': 'use back translation to generate more data', 'task-specific feature engineering': 'none'}.

Intro for the answer
> An example for this part of the prompt is as below:
> Here is the Python dictionary:

Figure 25: Prompt for data preparation module planning

The common instruction prompt

The textual task description from the user

Classification of the ML task extracted from the response from the task classification step

Your task is to suggest some pretrained models for the machine learning task. Some machine learning tasks only need one model, some tasks need multiple models (e.g., one for backbone and one for head, one for backbone and one for the framework, one for encoder and one for decoder, or one for generator and one for discriminator), and some tasks can be solved by either one model or multiple models.

> **Formatting instructions for the answer**
>> An example for this part of the prompt is as follows:
>> Your answer should be a Python list and only a python list. Each entry of the list is a dictionary that describes a choice of a single model or a combination of models.
>> For a choice of a single model, the dictionary should be a dictionary of a single key with the format of {name_of_model: explanation_of_the_suitability_of_model}. For a choice of a combination of models, the dictionary should be a dictionary of multiple keys with the format of {'name_of_model_1_in_the_combination+name_of_model_2_in_the_combination+...': {'reason:','explanation_of_the_suitability_of_the_combination'},{'name_of_model_1_in_the_combination': 'role_of_model_1_in_the_combination', 'name_of_model_2_in_the_combination': 'role_of_model_2_in_the_combination', ...}}. Examples of role_of_model_N_in_the_combination include 'backbone', 'head', 'encoder', 'decoder', 'generator', 'discriminator', and 'framework'.
>> The list can consist of only choices of a single model, only choices of combinations of models, or choices of both single model and combinations of models.
>> You should suggest [an integer from the settings] choices and the Python list should have [an integer from the settings] keys.

> **Intro for the answer**
>> An example for this part of the prompt is as below:
>> Here is the Python list:

Figure 26: Prompt for modeling module planning

The common instruction prompt

The textual task description from the user

Classification of the ML task extracted from the response from the task classification step

Your task is to generate the Python code for the data preparation module of the machine learning task.

Requirements of the code
An example for this part of the prompt is as below:
The generated Python code should meet the following requirements:
(1): There is a function named 'generate_dataloader' in the code.
(2): The data preparation module should load the files. All available files are listed in the user's description of the available files.
(3): The data preparation module should contain the following steps in a suitable order:
Plans for the data preparation module extracted from the response from the data preparation module planning step
(4): The returned objects of the function must be (train_loader,val_loader). train_loader is a torch.utils.data.DataLoader object that generates the training data and val_loader is a torch.utils.data.DataLoader object that generates the validation data. Both torch.utils.data.DataLoader objects should be an iterable that generates a batch of data in the format of a tuple of tensors. Each tensor in the tuple represents one type of tensor.
(5): The input and output data from the torch.utils.data.DataLoader objects must satisfy the following requirements:
Format of the data flow extracted from the response from the task classification step

Intro for the answer
An example for this part of the prompt is as below:
Here is the Python code:

Figure 27: Prompt for data preparation module code generation

The common instruction prompt

The textual task description from the user

Classification of the ML task extracted from the response from the task classification step

Your task is to generate the Python code for the modeling module of the machine learning task.

Requirements of the code
       An example for this part of the prompt is as below:
       The chosen model for the machine learning task is:
       Plans for the modeling module extracted from the response from the modeling module planning step
       The generated Python code should meet the following requirements:
       (1): There is a function named 'generate_model' in the code.
       (2): The returned values of the function must be a torch.nn.Module object.
       (3): The torch.nn.Module object should have a forward method that takes all types of tensor in input data as arguments. The value of each argument is a single-type tensor for a batch of data points. The forward method should return all types of tensors in output data. The value of each returned object is a single-type tensor for a batch of data points.
       (4): The input data and (ground-truth) output data are from a torch.utils.data.dataloader.DataLoader object. The input and output data from the torch.utils.data.dataloader.DataLoader object are already processed by data preparation steps (including tokenization for natural language processing tasks). The content of the processed input and output data is as follows:
       Format of the data flow extracted from the response from the task classification step

Intro for the answer
       An example for this part of the prompt is as below:
       Here is the Python code:

Figure 28: Prompt for modeling module code generation

There is a machine learning task, your goal is to generate the code for the evaluation metrics and for processing the output data for the evaluation metrics.

The common instruction prompt without the first two sentences

The textual task description from the user

Classification of the ML task extracted from the response from the task classification step

Format of the data flow extracted from the response from the task classification step

---

Requirements of the code
An example for this part of the prompt is as below:
The generated Python code should meet the following requirements:
(1): There is a function named 'generate_evaluation' in the code.
(2): The function 'generate_evaluation' should always include and only include the following arguments: An argument named 'predicted_scores'. The value of the argument is the torch.Tensor of predicted output about scores. An argument named 'actual_scores'. The value of the argument is the torch.Tensor of actual (ground truth) output about scores.
(3): The function 'generate_evaluation' should convert the raw output into the format acceptable by the evaluation metrics.
(4): The function 'generate_evaluation' should compare the predicted output(s) and actual output(s) based on the specified evaluation metrics in this machine learning task.
(5): The function 'generate_evaluation' should return one object, which is the numerical score based on the evaluation metrics in the format of Python built-in float.

---

Intro for the answer
An example for this part of the prompt is as below:
Here is the Python code:

---

Figure 29: Prompt for post processing module code generation

# F   Additional results

Table 6: Mean number of attempts per success with GPT-3.5.

| Datasets | CoT | Zero-shot | Reflexion | Self-Revision | Contextual Modular Generation (ours) | | |
|---|---|---|---|---|---|---|---|
| | | | | | w/o tests | w/o reflection | standard |
| Boston | $9.0 \pm 0.7$ | - | $5.5 \pm 0.8$ | $6.6 \pm 2.1$ | $3.7 \pm 0.5$ | $3.6 \pm 0.6$ | $\mathbf{3.1 \pm 0.6}$ |
| Iris | $\mathbf{3.0 \pm 0.2}$ | $3.4 \pm 0.6$ | $5.0 \pm 1.5$ | $5.0 \pm 1.0$ | $3.6 \pm 0.6$ | $3.2 \pm 0.4$ | $3.3 \pm 0.5$ |
| *Age* | - | - | $16.1 \pm 2.0$ | $20.3 \pm 6.8$ | - | $8.7 \pm 1.4$ | $\mathbf{8.3 \pm 1.9}$ |
| *Express* | - | - | $30.2 \pm 4.0$ | $17.3 \pm 3.2$ | - | $13.2 \pm 2.1$ | $\mathbf{11.0 \pm 2.6}$ |
| CIFAR-10 | $5.7 \pm 0.9$ | $6.7 \pm 1.3$ | $6.4 \pm 0.5$ | $20.2 \pm 5.4$ | - | $5.9 \pm 1.6$ | $\mathbf{5.6 \pm 0.6}$ |
| CIFAR-100 | $7.4 \pm 1.9$ | $7.6 \pm 1.3$ | $13.5 \pm 3.7$ | - | - | $7.5 \pm 1.3$ | $\mathbf{7.2 \pm 0.7}$ |
| *Whale* | - | - | - | - | - | $\mathbf{16.8 \pm 1.9}$ | $17.8 \pm 1.6$ |
| *Strip* | - | - | - | - | - | $\mathbf{13.0 \pm 2.8}$ | $14.7 \pm 2.9$ |
| IMDb Reviews | $6.2 \pm 0.7$ | $5.3 \pm 1.5$ | $6.4 \pm 1.3$ | $11.6 \pm 1.9$ | - | *3.6 ± 0.7* | $4.0 \pm 0.6$ |
| AG News | $11.8 \pm 2.2$ | $12.3 \pm 2.9$ | $10.4 \pm 1.5$ | - | - | $\mathbf{4.0 \pm 0.8}$ | $4.2 \pm 0.5$ |
| *Exam* | - | - | - | - | - | $15.0 \pm 5.8$ | $\mathbf{13.8 \pm 4.8}$ |
| *Learning* | - | - | - | - | - | $\mathbf{15.6 \pm 5.3}$ | $17.0 \pm 2.8$ |

Table 7: Mean number of attempts per success with PaLM 2.

| Datasets | CoT | Zero-shot | Reflexion | Self-Revision | Contextual Modular Generation (ours) | | |
|---|---|---|---|---|---|---|---|
| | | | | | w/o tests | w/o reflection | standard |
| Boston | $15.6 \pm 3.5$ | - | $8.3 \pm 1.8$ | $10.4 \pm 2.7$ | $4.4 \pm 0.7$ | $5.2 \pm 1.6$ | $\mathbf{3.4 \pm 0.5}$ |
| Iris | $3.9 \pm 0.7$ | $3.5 \pm 0.7$ | $5.3 \pm 1.3$ | $7.8 \pm 2.2$ | $3.8 \pm 0.7$ | $\mathbf{3.3 \pm 0.5}$ | $3.6 \pm 0.7$ |
| *Age* | - | - | - | - | - | $\mathbf{7.2 \pm 0.6}$ | $7.9 \pm 1.5$ |
| *Express* | - | - | - | - | - | $20.2 \pm 5.2$ | $\mathbf{15 \pm 1.7}$ |
| CIFAR-10 | $8.7 \pm 0.8$ | - | $9.8 \pm 4.0$ | - | - | $7.4 \pm 0.7$ | $\mathbf{7.3 \pm 1.0}$ |
| CIFAR-100 | $\mathbf{7.4 \pm 1.3}$ | $20.2 \pm 5.1$ | $16.8 \pm 3.3$ | - | - | $8.8 \pm 1.4$ | $7.4 \pm 1.8$ |
| *Whale* | - | - | - | - | - | $18.3 \pm 7.3$ | $\mathbf{18.0 \pm 3.0}$ |
| *Strip* | - | - | - | - | - | - | $\mathbf{33.3 \pm 7.6}$ |
| IMDb Reviews | $5.8 \pm 1.2$ | $7.9 \pm 1.0$ | $10.5 \pm 2.1$ | $12.2 \pm 1.1$ | - | $5.0 \pm 0.5$ | $\mathbf{3.9 \pm 0.5}$ |
| AG News | $12.2 \pm 1.0$ | - | $18.2 \pm 3.4$ | - | - | $6.0 \pm 1.1$ | $\mathbf{4.9 \pm 0.7}$ |
| *Exam* | - | - | - | - | - | $17.7 \pm 2.5$ | $\mathbf{14.6 \pm 4.9}$ |
| *Learning* | - | - | - | - | - | $19.8 \pm 2.4$ | $\mathbf{16.7 \pm 4.3}$ |

Table 8: Mean iterations to correct errors. (1) tests-only: feedback is provided solely through unit tests (Figure 8 and Algorithm 3). (2) reflection: feedback is provided through both unit tests and the self-reflection mechanism.

| Modules | GPT-4 | | GPT-3.5 | | PaLM 2 | |
|---|---|---|---|---|---|---|
| | tests-only | reflection | tests-only | reflection | tests-only | reflection |
| Data preparation | $\mathbf{4.1 \pm 1.8}$ | $4.2 \pm 1.9$ | $5.6 \pm 2.1$ | $6.0 \pm 1.7$ | $5.8 \pm 2.4$ | $6.1 \pm 2.2$ |
| Modeling | $2.7 \pm 1.1$ | $\mathbf{2.4 \pm 1.1}$ | $2.9 \pm 1.4$ | $3.3 \pm 1.9$ | $5.4 \pm 2.2$ | $5.4 \pm 3.6$ |
| Post processing | $3.0 \pm 1.0$ | $\mathbf{2.8 \pm 1.4}$ | $3.1 \pm 1.9$ | $2.7 \pm 1.2$ | $4.5 \pm 1.5$ | $3.9 \pm 1.2$ |

