# OpenReview forum: "Large Language Models Synergize with Automated Machine Learning"
_TMLR — Accepted by TMLR_

### Review · Reviewer_gNyJ · 2024-06-24

**Summary Of Contributions:**

This paper presents a new method to synthesize machine learning programs using
large language models and automated machine learning. The authors derive a program
decomposition method that breaks down the task into smaller modules, and then utilize unit tests
to guide the search for the best combination of modules. This method
automates the entire ML workflow, breaking it into three steps: data preprocessing, modelling, and post-processing.
The authors demonstrate the effectiveness of their method on a small set of tasks, showing that it
outperforms common baselines in neural program synthesis such as chain-of-thought and and Reflexion.

**Audience:**

Yes

**Claims And Evidence:**

Yes

**Requested Changes:**

- The authors should provide a more detailed evaluation of the individual components of their method. This would help to understand the contribution of each component to the overall performance of the method.

**Strengths And Weaknesses:**

## Strengths

- The method is interesting and the authors provide a detailed explanation of the approach and the rationale behind components of the method.
- A theoretical analysis of their approach is provided, which is quite welcome as it is often missing in similar works.

## Weaknesses

- This method includes a large number of components which, despite being justified in text, are not evaluated in isolation.
  This makes it difficult to understand the contribution of each component to the overall performance of the method.

---

> ### Author Response · Authors · 2024-07-18
> **Detailed evaluation results**
>
> Thank you very much for your time in reviewing our work and we thank you for your feedbacks!
>
> We provide our response below:
>
> **“This method includes a large number of components which, despite being justified in text, are not evaluated in isolation. This makes it difficult to understand the contribution of each component to the overall performance of the method.”**
>
> **"The authors should provide a more detailed evaluation of the individual components of their method. This would help to understand the contribution of each component to the overall performance of the method."**
>
> Thanks for the comments.
>
> - Our method comprises three phases: code generation, code testing, and code optimization (Figure 3 in the paper).
> - In the code generation phase, we introduce a novel sequence generation method called Contextual Modular Generation, which includes three key mechanisms: (1) breaking down the program, (2) testing the generated code, and (3) reflecting on previous sequence generation history.
>
> **Changes we made: We added the experimental results about the mean number of iterations to correct the code with errors under the scenario where only mechanisms (1) and (2) are activated vs the scenario where all three mechanisms are activated. We added the results in the last paragraph in Section 6.2 and Table 8 in Appendix F of the paper.**
>
> - This result provides insights into the effectiveness of mechanism (2) and mechanism (3) in the code generation phase from a different perspective. In addition, the result also offers insights about the code testing phase, since mechanism (2) is about testing the generated code with unit tests.
> - Tables 1, 6, and 7 in the paper present the results for various combinations of these three mechanisms. The column "w/o reflection" represents the scenario where mechanism (3) is removed. The column "w/o tests" shows the results when both mechanisms (2) and (3) are removed. Since mechanism (3) relies on mechanism (2), a combination of mechanisms (1) and (3) without (2) is not possible. The column "Reflexion" indicates the scenario where mechanism (1) is removed. Therefore, Tables 1, 6, and 7 in the paper contain ablation studies for the code generation phase.
> - Similarly, for the code optimization phase, the results also contain ablation studies. Table 3 in the paper shows the results of the baseline (No Search), with the search algorithm but without ZC proxies (Search only), and with both the search algorithm and ZC proxies (Search + ZC proxies).

---

### Review · Reviewer_br1n · 2024-07-01

**Summary Of Contributions:**

**Integration of LLMs with AutoML**:  The study explores the synergistic relationship between LLMs and AutoML, proposing a framework where LLMs generate machine learning (ML) programs that AutoML systems can then optimize. This integration addresses the limitations of current AutoML methods, which often require substantial human expertise for programming and are resource-intensive.

**Text-to-ML Framework**: The paper introduces a novel Text-to-ML framework that automates the entire ML program synthesis process. This framework converts textual task descriptions into optimized ML programs through a multi-step sequence-to-sequence (seq2seq) approach. Each step involves generating and optimizing program components using LLMs and AutoML.

**Enhanced Sequence Generation**: By incorporating a self-reflection mechanism, the proposed methods improve the generation of valid ML programs. The self-reflection mechanism allows the system to retain feedback from previous trials, enhancing the accuracy and efficiency of subsequent generations.

**Robust Evaluation**: The paper conducts extensive experiments using various datasets and compares different sequence generation methods. The results demonstrate that the proposed methods, particularly those incorporating self-reflection, outperform others in generating valid ML programs.

**Audience:**

Yes

**Broader Impact Concerns:**

The democratization of machine learning (ML) through the proposed Text-to-ML framework increases accessibility, which can be both a benefit and a risk. While it empowers non-experts to leverage ML, it also raises the potential for misuse by individuals with malicious intent or without adequate understanding of ML principles. A Broader Impact Statement should address the measures in place to prevent misuse, such as guidelines for responsible use and potential monitoring mechanisms.

The automation of ML program synthesis can obscure the decision-making process, making it difficult to understand how certain outcomes are derived. This lack of transparency can hinder accountability, especially in critical applications like healthcare or criminal justice. The Broader Impact Statement should emphasize the importance of transparency and accountability, suggesting ways to ensure that automated processes are interpretable and that outcomes can be traced back to specific decisions.

**Claims And Evidence:**

Yes

**Requested Changes:**

The current framework may not be fully capable of addressing highly complex ML tasks. Enhancing the framework to better handle these tasks through advanced optimization techniques and more sophisticated models is crucial.

The implementation's efficiency, particularly in practical applications, needs improvement. Optimizing the sequence generation process and reducing computational requirements are necessary steps.

Developing methods to better compare automated solutions with human-generated ones would provide valuable insights into the performance gap and areas for improvement. This comparison is essential to demonstrate the framework's effectiveness in real-world scenarios.

**Strengths And Weaknesses:**

Strengths

   - The paper introduces an approach by integrating Large Language Models (LLMs) with AutoML systems. This synergy addresses the limitations of current AutoML methods and showcases the potential for enhanced machine learning automation.
   - The paper provides extensive experimental results, comparing different sequence generation methods across various datasets. This robust evaluation demonstrates the effectiveness of the proposed methods, particularly those incorporating self-reflection.

Weaknesses

   - While the framework shows promise for various tasks, it may struggle with highly complex ML tasks. The paper acknowledges this limitation, indicating a need for further research to address such challenges. The paper notes the infeasibility of comparing automated solutions with human-generated ones in competitive environments like Kaggle.
   - The scalability and generalization of the proposed methods to different domains and datasets may require further validation. Ensuring that the framework can handle diverse scenarios is crucial for broader applicability.
   - The integration of LLMs with AutoML systems might be resource-intensive, potentially limiting its use in resource-constrained environments. Addressing this issue would be important for wider adoption.

---

> ### Author Response · Authors · 2024-07-18
> **Focus of the study vs future directions**
>
> Thank you very much for your time on reviewing our work and we thank you for your feedbacks!
>
> We provide our response below:
>
> **1. “While the framework shows promise for various tasks, it may struggle with highly complex ML tasks. The paper acknowledges this limitation, indicating a need for further research to address such challenges. The paper notes the infeasibility of comparing automated solutions with human-generated ones in competitive environments like Kaggle.”**
>
> **2. “The current framework may not be fully capable of addressing highly complex ML tasks. Enhancing the framework to better handle these tasks through advanced optimization techniques and more sophisticated models is crucial.”**
>
> **3. “Developing methods to better compare automated solutions with human-generated ones would provide valuable insights into the performance gap and areas for improvement. This comparison is essential to demonstrate the framework's effectiveness in real-world scenarios.”**
>
> Thanks for the comments.
>
> **Changes we made:**
>
> **1. In Section 7.3 of the paper, we added a discussion (the 4th paragraph) to address how to overcome such limits.**
>
> **2. We add the details of the GPT versions in Section 6.1 of the paper. We additionally add the details regarding the prompts of the intermediate step in Appendix E of the paper. The added implementation details would help future studies to implement and extend our method with additional techniques.**
>
> - If we understood the comments correctly, Reviewer br1n means overcoming the limitations will be important for future research.
> - We agree that the limitations, as already stated in Section 7.3 of the paper, including struggling with highly complex ML tasks and infeasibility of comparison with human-generated solutions, are important aspects that research on this topic should address.
> - We think such limitations should be mostly handled in future research. This study is mainly an exploration of a novel paradigm of solving ML tasks: from text to ML programs. We mainly explore whether it is feasible to automatically generate and optimize ML programs from just texts. In addition, practically speaking, such an exploration also shows promises for regular users under limited computational resources.
> - The medal-awarding ML tasks at the Kaggle platform, even the simpler ones, are already very challenging.
> - The goal of automation of ML tasks at the code level might not conflict with the goal of performance enhancement for experts in a competitive environment.
> - The added parts in Section 7.3 discuss future studies that are more geared towards the second goal, including techniques that may achieve this goal. While our method can be extended with those techniques, such techniques will add many more components to this study. This study already incorporates many components, from code generation, code testing, to code optimization. We want to avoid excessive complexity in just one paper. Therefore, we think such a goal would be better addressed in future works.

---

> ### Author Response · Authors · 2024-07-18
> **The scalability and generalization of the methods**
>
> **“The scalability and generalization of the proposed methods to different domains and datasets may require further validation. Ensuring that the framework can handle diverse scenarios is crucial for broader applicability.”**
>
> Thanks for this comment.
>
> **Changes we made: We added in paragraph 1 of Section 7.3 that the practical significance of the theoretical linear complexity should be more extensively tested for other types of sequence generation tasks.**
>
> - We acknowledge such limitations regarding the experimental verification of the theoretical results.
> - We think future research might be better suited to extensively test the linear scalability in a more general scenario. Such experiments might include other types of sequence generation tasks. Although, as stated in Section 3.1 of the paper, the sequence generation method is generic, as long as the sequence evaluation method defined in Section 3.2 (Definition 1) is available. We did not perform experiments to test its performance in a generic scenario. The focus of our study is on synthesizing the program for ML tasks rather than the general applicability of a sequence generation approach.

---

> ### Author Response · Authors · 2024-07-18
> **Resource consumption of the method**
>
> **1. “The integration of LLMs with AutoML systems might be resource-intensive, potentially limiting its use in resource-constrained environments. Addressing this issue would be important for wider adoption.”**
>
> **2. “The implementation's efficiency, particularly in practical applications, needs improvement. Optimizing the sequence generation process and reducing computational requirements are necessary steps.”**
>
> Thanks for the comments.
>
> - Our method consists of three phases: code generation, code testing, and code optimization.
> - Accurate measurement of the time the code generation phase of our method takes is infeasible, as the vast majority amount of the time is spent waiting for the response from the LLMs. The response time from the LLMs (GPT-4 and GPT 3.5) varies a lot depending on the server conditions. However, a rough estimate is that with GPT-4, about 50 different valid candidate ML programs can be generated in 15 minutes. With GPT-3.5, the process is generally multiple times faster. Such an efficiency is much higher than the average speed of writing ML programs by human programmers, not to mention the process is fully automatic.
> - The time the code testing phase takes is small compared to the other two phases because the testing phase relies on simulated data, which only consists of a single batch of datapoints.
> - The time the code optimization phase takes can be much shorter than naive manual optimization approaches. A ratio (50% in our experiments) of the least-promising models based on the estimation of the zero-cost (ZC) proxies is filtered out. The ZC proxy evaluations of a single model take no more than a fraction of a second to at most several seconds, which is negligible compared to fully training and testing the model.
> - We agree that an important future direction would be to further accelerate and simplify the process for wider adoption.

---

> ### Author Response · Authors · 2024-07-18
> **Broader impact concerns**
>
> **1. “The democratization of machine learning (ML) through the proposed Text-to-ML framework increases accessibility, which can be both a benefit and a risk. While it empowers non-experts to leverage ML, it also raises the potential for misuse by individuals with malicious intent or without adequate understanding of ML principles. A Broader Impact Statement should address the measures in place to prevent misuse, such as guidelines for responsible use and potential monitoring mechanisms.”**
>
> **2. “The automation of ML program synthesis can obscure the decision-making process, making it difficult to understand how certain outcomes are derived. This lack of transparency can hinder accountability, especially in critical applications like healthcare or criminal justice. The Broader Impact Statement should emphasize the importance of transparency and accountability, suggesting ways to ensure that automated processes are interpretable and that outcomes can be traced back to specific decisions.”**
>
> Thanks for the comments.
>
> **Changes we made:**
>
> **1. We added a comprehensive broader impact statement at the end of the main text of the paper. The broader impact statement is divided into two sections, with each section addressing one of the comments.**
>
> **2. In the implementation of our method (code in the supplementary data), we improved the code to make the implementation also clearly display the prompts and responses in each step involving LLMs. Such a mechanism increases the transparency and accountability of the process.**
> - We are dedicated to the responsible use of AI, and we advocate for more explainable AI.
> - As pointed out in the broader impact statement, a notable feature of our implementation of Text-to-ML is that the implementation includes a comprehensive logger to record the automatic process, which also increases transparency and accountability.

---

### Review · Reviewer_TZVm · 2024-07-04

**Summary Of Contributions:**

This paper focuses on program synthesis from LLMs, and particularly for an autoML setting. Given text descriptions of the task, the authors propose a new prompting framework, contextual modular generation, to prompt the LLMs to generate the machine learning programs. The prompting framework is novel to integrate some design heuristics from software engineering, such as decomposing the program synthesis as generating several smaller modules respectively (mainly classified as data preparation, modeling, and postprocessing). The program of each module is verified through unit tests from LLMs to ensure different modules are compatible with each other and the entire program is free of syntax errors. This paper also presents a theoretical analysis on the scalability of the method over the length of complex output sequences, which indicates that the generation difficulty of contextual modular generation scales linearly over the problem length in the best case. Experimental results on 12 datasets demonstrate that the proposed method outperforms baselines such as COT and reflexion, identifying valid programs with the least number of attempts.

**Audience:**

Yes

**Broader Impact Concerns:**

The paper focuses on program synthesis for an autoML purpose, which is possible to produce ML programs that can automatically train a model to learn from toxic data for certain purposes. This aspect may warrant some discussion which the current version of the paper lacks.

**Claims And Evidence:**

Yes

**Requested Changes:**

1. I suggest to include specific prompts used to instruct the LLMs at each stage in the appendix, so that the readers can better understand what is going on exactly from the specific examples.
2. I suggest clarifying the GPT-4 versions that the paper use in the main text of the paper, given that there are so many versions of GPT-4 and the authors talk about the knowledge cutoff date.

**Strengths And Weaknesses:**

### Strengths

1. The proposed method is novel and integrates software engineering practices in a relatively neat way, specifically tailored for autoML.
2. The text-to-ML task is practically useful and challenging while the role of LLMs for it is not extensively studied yet.
3. This paper is comprehensive on addressing the details and includes theoretical analysis for the complexity of the proposed approach.
4. The empirical results are good on multiple benchmarks.

### Weaknesses

1. I feel some details are difficult to follow and it is not very clear how the LLMs are instructed exactly at each stage. In the appendix, I suggest the authors give examples for the exact prompt used for LLMs for each stage during the process. Currently the appendix only shows the general prompt during evaluation.
2. Several of the popular datasets such as CIFAR and IMDB review classification are widely used over the Internet. Thus the LLMs are very likely to have seen many programs for these datasets already.

---

> ### Author Response · Authors · 2024-07-18
> **Details in the methods**
>
> Thank you very much for your time in reviewing our work and we thank you for your feedbacks!
>
> We provide our response below:
>
> **1. “I feel some details are difficult to follow and it is not very clear how the LLMs are instructed exactly at each stage. In the appendix, I suggest the authors give examples for the exact prompt used for LLMs for each stage during the process. Currently the appendix only shows the general prompt during evaluation.”**
>
> **2. “I suggest to include specific prompts used to instruct the LLMs at each stage in the appendix, so that the readers can better understand what is going on exactly from the specific examples.”**
>
> **3. “I suggest clarifying the GPT-4 versions that the paper use in the main text of the paper, given that there are so many versions of GPT-4 and the authors talk about the knowledge cutoff date.”**
>
> Thanks for the comments.
>
> **Changes we made:**
>
> **1. We added such information about the GPT versions in Section 6.1 of the paper.**
>
> **2. We added a more detailed description of the prompts for the intermediate steps (rather than the user input) in Appendix E of the paper**
>
> **3. In the implementation of our method (code in the supplementary data), we improved the code to make the implementation also clearly display the prompts and responses in each step involving LLMs. Such a mechanism increases the transparency and accountability of the process.**
>
> - The GPT-4 version we used in our experiments is GPT-4-0613. In addition, the GPT-3 version is GPT-3.5-turbo-0613, both have a knowledge cutoff date of September 2021
> - As stated in Section 3.2, our sequence generation method, Contextual Modular Generation, incorporates a mechanism from the method Reflexion [1], which involves instruction constructors that assemble the prompt at each step based on the output sequences from previous steps. As a result, the prompts in the intermediate steps are not constant but rather dynamically and automatically constructed. Therefore, to understand the exact prompts in the intermediate steps, we suggest that the best option is to run our implementation and check the dynamically constructed prompts from the message output in action.
>
> References:
>
> 1.	Noah Shinn, Federico Cassano, Ashwin Gopinath, Karthik R Narasimhan, and Shunyu Yao. Reflexion: Language agents with verbal reinforcement learning. In the Thirty-seventh Conference on Neural Information Processing Systems, 2023.

---

> ### Author Response · Authors · 2024-07-18
> **Datasets and the training corpus of the LLMs**
>
> **“Several of the popular datasets such as CIFAR and IMDB review classification are widely used over the Internet. Thus the LLMs are very likely to have seen many programs for these datasets already.”**
>
> Thanks for this comment.
>
> - We agree with reviewer TZVm that information about these datasets probably frequently appears in the training data of the LLMs, as already stated in Section 6.1 of the paper
> - Thus, our datasets also include datasets from Kaggle, which are after the knowledge cutoff date of the LLMs (Appendix D.2)
> - In the experiments, we observed that existing methods, such as CoT and Zero-shot, could handle those common datasets, but they struggle with the Kaggle datasets, highlighting the advantages of our methods for datasets outside the LLM’s training corpus (Tables 1, 6, and 7)

---

> ### Author Response · Authors · 2024-07-18
> **Broader impact concerns**
>
> **“The paper focuses on program synthesis for an autoML purpose, which is possible to produce ML programs that can automatically train a model to learn from toxic data for certain purposes. This aspect may warrant some discussion which the current version of the paper lacks.”**
>
> Thanks for this comment.
>
> **Changes we made: We added a comprehensive broader impact statement at the end of the main text of the paper. Specifically, we discuss the need to avoid bias and toxicity in data when using Text-to-ML.**
> - We are dedicated to the responsible use of AI, and we advocate for more explainable AI.
> - We state that users should be informed of ethical principles in ML/AI before utilizing Text-to-ML

---

> > ### Comment · Reviewer_TZVm · 2024-07-21
> >
> > Thank you for the response and update!

---

### Author Response · Authors · 2024-07-17
**Global response**

We sincerely thank the action editor and all reviewers for their timely responses. Your insights and perspectives are invaluable for improving our paper. We appreciate the reviewers for highlighting both the strengths and weaknesses of our work. We just updated the paper and the supplementary data in response to the reviewer's comments.

We sincerely appreciate the time you have dedicated to reviewing our work. Your feedback has been invaluable, and we have taken your comments seriously, making necessary adjustments, including additional experiments, clarifications, and revisions to the draft.

Please take a moment to review our responses and let us know if you have any further questions or need additional information. We believe we have addressed all your concerns, but we remain open to further queries.

Thank you once again for your time and effort. We look forward to your response.

---

### Decision · Action_Editor_sRS4 · 2024-08-06

**Recommendation:** Accept as is

**Comment:**

Reviewers all recommended accepting this paper. There were initial concerns about lack of clarity that were resolved during the rebuttal, so that the current draft should be clear and readable. Reviewers did note that the proposed method was simple and that it could primarily target very simple ML applications, but level of novelty is not a criterion for acceptance at TMLR and many applciations of ML involve simple problems.

**Audience:**

Reviewers agree that this paper would be of interest primarily to individuals who aim to solve relatively simple Kaggle-style ML tasks. More broadly, one reviewer noted that "this paper would be interesting to the community". Therefore it is likely that some individuals from TMLR's audience would be interested and therefore it would be appropriate for TMLR.

**Claims And Evidence:**

Reviewers noted that "the paper is excellent in its extensive experiments and detailed discussions" and "their method is well evaluated, including ablations for specific components, showing the effectiveness of each part of the system", so the claims and evidence are sound.